# Evolutionary recruitment of flexible *Esrp*-dependent splicing programs into diverse embryonic morphogenetic processes

Demian Burguera[1,2,3], Yamile Marquez[1,3], Claudia Racioppi[4,5], Jon Permanyer[1,3], Antonio Torres-Méndez [1,3], Rosaria Esposito[4], Beatriz Albuixech-Crespo[2], Lucía Fanlo[6], Ylenia D'Agostino[4], Andre Gohr[1,3], Enrique Navas-Perez[2], Ana Riesgo[7], Claudia Cuomo[4], Giovanna Benvenuto [4], Lionel A. Christiaen [5], Elisa Martí[6], Salvatore D'Aniello[4], Antonietta Spagnuolo[4], Filomena Ristoratore [4], Maria Ina Arnone[4], Jordi Garcia-Fernàndez [2] & Manuel Irimia [1,3]

Epithelial-mesenchymal interactions are crucial for the development of numerous animal structures. Thus, unraveling how molecular tools are recruited in different lineages to control interplays between these tissues is key to understanding morphogenetic evolution. Here, we study *Esrp* genes, which regulate extensive splicing programs and are essential for mammalian organogenesis. We find that *Esrp* homologs have been independently recruited for the development of multiple structures across deuterostomes. Although *Esrp* is involved in a wide variety of ontogenetic processes, our results suggest ancient roles in non-neural ectoderm and regulating specific mesenchymal-to-epithelial transitions in deuterostome ancestors. However, consistent with the extensive rewiring of *Esrp*-dependent splicing programs between phyla, most developmental defects observed in vertebrate mutants are related to other types of morphogenetic processes. This is likely connected to the origin of an event in *Fgfr*, which was recruited as an *Esrp* target in stem chordates and subsequently co-opted into the development of many novel traits in vertebrates.

[1] Centre for Genomic Regulation (CRG), Barcelona Institute of Science and Technology (BIST), Dr Aiguader 88, Barcelona 08003, Spain. [2] Department of Genetics, School of Biology, and Institut de Biomedicina (IBUB), University of Barcelona, Diagonal 643, Barcelona 08028, Spain. [3] Universitat Pompeu Fabra (UPF), Barcelona 08003, Spain. [4] Stazione Zoologica Anton Dohrn, Villa Comunale, 80121 Napoli, Italy. [5] Center for Developmental Genetics, Department of Biology, New York University, New York, NY 1003, USA. [6] Instituto de Biología Molecular de Barcelona, CSIC, Parc Científic de Barcelona, Baldiri Reixac 20, Barcelona 08028, Spain. [7] Department of Life Sciences, Natural History Museum of London, Cromwell Road, SW7 5BD London, UK. Yamile Marquez and Claudia Racioppi contributed equally to this work. Correspondence and requests for materials should be addressed to M.I.A. (email: miarnone@szn.it) or to J.G.-F. (email: jordigarcia@ub.edu) or to M.I. (email: mirimia@gmail.com)

During embryo development, tissues proliferate and differentiate in a coordinated manner to build a whole organism through a genome-guided process. Different cell types express distinct transcriptomes to control cellular identity and physiology, and to establish differential interaction capabilities between embryonic tissues. Final morphology is thus achieved by cell-specific transcriptomic responses to external and internal stimuli within each tissue. Therefore, changes in the genetic networks involved in morphogenesis are ultimately responsible for both the modification of organs and, at a macroevolutionary scale, the origin of new structures[1,2].

In particular, epithelial-mesenchymal interplays are essential to many organogenetic processes in vertebrates[3,4]. These tissues often interact in morphogenetic interfaces through the exchange of cells and signaling molecules[5,6]. Despite the great diversity of cell types across the embryo, the majority can be classified as showing either mesenchymal or epithelial characteristics. This broad distinction, which is independent of tissue origin, has also been shown to be reflected in the patterns of gene expression and alternative splicing (AS)[7]. Those transcriptomic programs confer partly antagonistic morphogenetic properties to epithelial and mesenchymal tissues by modulating certain cellular features, such as adhesion, motility and polarity.

Of particular importance for mammalian morphogenesis is a mutually exclusive exon skipping event found in members of the FGF receptor (*Fgfr*) gene family[8]. Exons IIIb and IIIc encode a region of the third immunoglobulin domain (IgIII) of the FGFR1, FGFR2, and FGFR3 proteins, and are differentially included in transcripts from epithelial or mesenchymal cells, respectively. Importantly, their mutually exclusive inclusion has a dramatic effect on the affinity of the receptors for FGF ligands[9], providing epithelial cells specificity for FGF signals secreted by the mesenchyme, and vice versa. Consistent with the importance of this regulatory system in development, disruption of the Fgfr2-IIIb isoform leads to severe defects during mice organogenesis[10].

These and other morphogenesis-associated AS events are directly regulated by the *Epithelial Splicing Regulatory Protein* (*Esrp*) genes in mammalian species[11]. *Esrp1* and *Esrp2* were originally identified as positive regulators of IIIb exon inclusion of the *Fgfr2* gene[12]. They encode RNA-binding proteins that are dynamically expressed mainly in a subset of epithelial tissues during mouse development[13], although mesenchymal expression has also been reported in chicken[14]. Recently, double knockout (DKO) mice for both *Esrp* genes were shown to display severe organogenetic defects and a complete shift to exon IIIc inclusion in *Fgfr1*, *Fgfr2*, and *Fgfr3*[15]. In addition, many *Esrp* exon targets were identified in genes involved in cell–cell adhesion, cell polarity, and migration[16]. However, the origin and evolution of *Esrp* morphogenetic functions and its regulated AS programs remain largely unknown.

Here, to investigate the evolution of *Esrp* functions and associated transcriptomic programs, we performed *Esrp* loss-of-function or gain-of-function experiments in several deuterostome species. Within bony vertebrates, *Esrp* genes play conserved roles in the development of numerous homologous organs. Consistently, *Esrp* regulates a core set of homologous exons in the three studied vertebrate species, including the mutually exclusive exons in the *Fgfr* family. Study of three non-vertebrate deuterostomes showed that *Esrp* is involved in a wide variety of morphogenetic processes in multiple unrelated structures in these species, and that, among others, it likely played an ancestral role in regulating specific mesenchymal-to-epithelial transitions (METs) in the deuterostome ancestor. However, transcriptomic analyses showed that most exons present clade-restricted differential regulation. In particular, no *Esrp*-dependent alternative exons were found conserved between studied species belonging to

different phyla. Exemplifying this split, we show that the *Fgfr* event affecting the IgIII domain originated from a Bilateria hotspot of recurrent AS evolution that was co-opted as an *Esrp* target at the base of chordates.

## Results

**esrp1 and esrp2 are involved in morphogenesis in zebrafish.** A broad phylogenetic analysis showed that the *Esrp* family predates the origin of metazoans and that a single copy of *Esrp* has been maintained in most metazoan groups with the exception of the vertebrate lineage, in which two copies are present in all studied species (Supplementary Figs. 1, 2). To investigate the evolution of *Esrp* roles across deuterostomes at various phylogenetic distances, we first examined the expression and function of the two *Esrp* paralogs (*esrp1* and *esrp2*) in zebrafish. A highly dynamic expression pattern was observed for both genes during the development of zebrafish using whole-mount in situ hybridization (WMISH) (Fig. 1a, Supplementary Fig. 3). At early stages, both genes displayed different expression patterns. *esrp1* transcripts were detected in the whole epidermis at 14 h post-fertilization (h.p.f.), but its expression was restricted to the posterior part of the embryo at 16 h.p.f.. On the other hand, *esrp2* was found only in the hatching gland rudiment at these stages. However, from 24 h.p.f. to 5 days post-fertilization (d.p.f.), the expression of both genes converged in most territories. Their transcription was transiently activated during the development of multiple organs, including the olfactory epithelium, otic vesicle, pharynx, epidermis and notochord. *esrp2* showed expression in a few additional tissues, such as pronephros, hatching gland, liver, and heart.

Next, we used the CRISPR-Cas9 system to generate loss-of-function zebrafish mutant lines for *esrp1* and *esrp2*. We targeted single guide-RNAs to the first and third exons of *esrp1* and *esrp2*, respectively (Fig. 1b). For *esrp1*, we selected a mutant allele with a 168-bp deletion and 14-bp insertion that induced the usage of an upstream cryptic splice donor, resulting in skipping of the whole coding sequence of exon 1, including the start codon. For *esrp2*, we selected a mutant allele with a 17-bp frame-disrupting deletion that produced a premature termination codon before the translation of any RNA-binding domain and that was predicted to trigger non-sense mediated decay (Fig. 1b). Expression of the mutant alleles was highly reduced in homozygous embryos, while transcript levels of the other paralog were not increased as a compensatory mechanism in the single mutants, as shown by quantitative PCR (Supplementary Fig. 3m). Furthermore, western blot assays confirmed the loss of the full-length protein for both mutant alleles, although faint bands that might correspond to shorter truncated proteins or unspecific signal were detected for *esrp1* (Supplementary Fig. 3n). Therefore, to further determine the functional impact of the mutations, we cloned the mutant and wild-type (WT) *esrp1* alleles in a pcDNA3.1 vector and transfect the construct into human 293T cells. Whereas expression of the WT allele of zebrafish *esrp1* was able to modify the splicing pattern of previously reported endogenous targets (*EXOC7*, *ARHGAP17*, and *FGFR1*) in the same way as its human ortholog[16], expression of the mutant *esrp1* allele did not produce any measurable effect in exon inclusion levels (Supplementary Fig. 3o), although an effect on other regulatory processes cannot be ruled out. We were unable to clone the full-length transcript of the mutant *esrp2* allele due to its reduced expression. Altogether, these data support an efficient loss of function for both *Esrp* genes in our mutant lines, particularly with regards to splicing regulation.

Morphological examination of single *esrp1* or *esrp2* homozygous zebrafish mutants showed no apparent gross defects

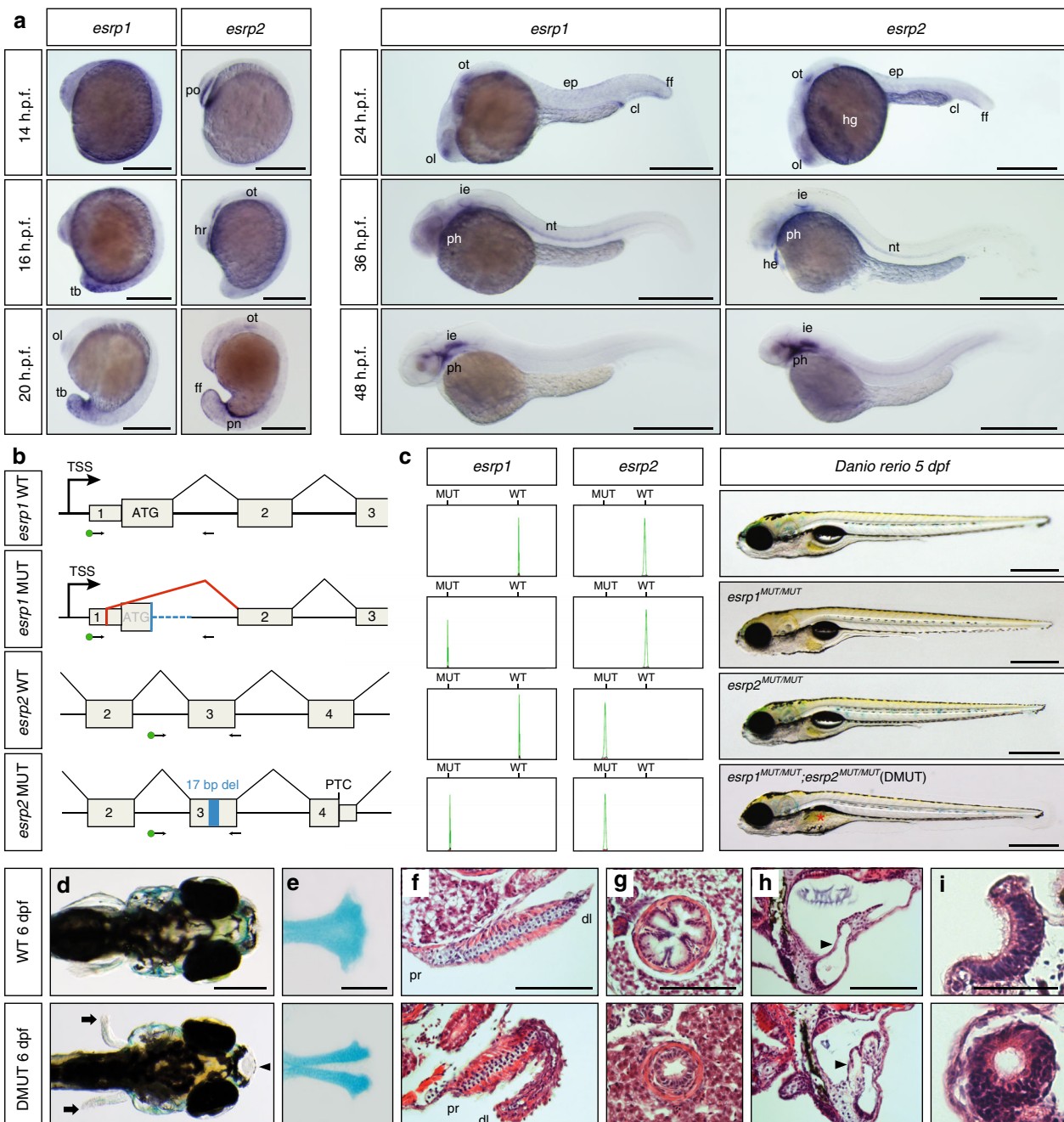

**Fig. 1** Expression and developmental roles of *esrp1* and *esrp2* in zebrafish. **a** WMISH for *esrp1* and *esrp2* in *Danio rerio* WT embryos. At 14 h.p.f., *esrp1* transcripts were observed in embryonic epidermis, while *esrp2* expression was only detected in the polster (po). At 16 h.p.f., *esrp1* was restricted to the posterior and tailbud (tb) epidermis, whereas *esrp2* persisted in the hatching gland rudiment (hr) and mild expression started to be detected in the otic placode (ot). By 20 h.p.f., *esrp1* was found in the tailbud epidermis and more subtly in the olfactory placode (ol), while *esrp2* appeared in new territories, such as pronephros (pn) and ectodermal cells of tailbud fin fold (ff). At 24 h.p.f., expression of both paralogs presented a similar pattern including olfactory and otic placodes, cloaca (cl), and epidermis (ep), although *esrp2* was also observed in the hatching gland (hg). By 36 h.p.f., both genes were detected in the inner ear epithelium (ie), notochord (nt), and phanynx (ph), and *esrp2* was also observed in the heart (he). At 48 h.p.f., expression was found predominantly in inner ear and pharynx. **b** Schematic representation of the genomic and transcriptomic impact of the selected *esrp1* and *esrp2* mutations. Blue boxes/lines represent genomic deletions in the mutants, while the red line depicts an altered splice junction in the *esrp1* mutant allele. TSS, transcription start site; PTC, premature termination codon; del, deletion. Standard and fluorescent (green dot) primers used during genotyping are represented by arrows. **c** Left: genotyping of embryos by fluorescent PCR readily distinguished between WT and MUT alleles. Right: Representative 5 d.p.f. larvae for wild type (WT), *esrp1* mutant (*esrp1^MUT/MUT^*), *esrp2* mutant (*esrp2^MUT/MUT^*), and double mutant (DMUT) genotypes. Deflated swim bladder in the DMUT embryo is indicated by a red asterisk. **d–i** Phenotypic differences between 6 d.p.f. WT (top) and DMUT (bottom) embryos in different embryonic structures. DMUT larvae showed impaired fin formation (arrows) and cleft palate (arrowhead) **d**, including malformation of the ethmoid bone, as shown by Alcian blue staining **e**. **f–i** Transversal histological sections stained with hematoxylin and eosin showing structural differences in pectoral fin **f**, esophagus **g**, inner ear **h** and olfactory epithelium **i**. Black arrowheads mark the dorso-lateral septum between semicircular canals in **h**. Proximal (pr) and distal (dl) parts of the fin are indicated in **f**. Scale bars: 1 mm **a**, 2 mm **c–e**, 100 µm **f–h**, 50 µm **i**

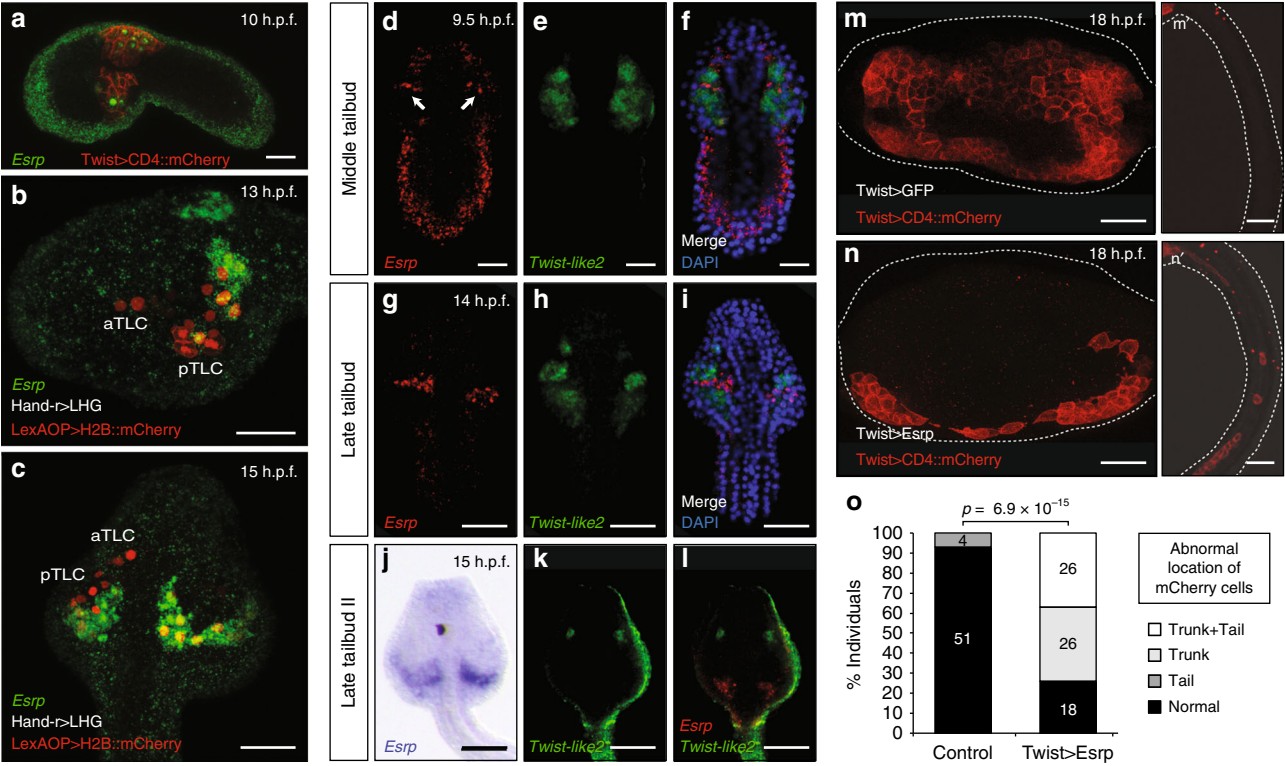

**Fig. 2** *Esrp* is expressed in a subpopulation of TLCs in *Ciona* and is able to modulate cell motility in the mesenchymal lineage. **a** *Esrp* expression (green) is detected at 10 h.p.f. by fluorescent WMISH in the epidermis and in some cells within the mesenchymal lineage, as shown by co-staining with mChe driven by the Twist promoter, which labels the *Twist-like1*-derived cell lineage. **b**, **c** *Esrp* (green) is expressed in the TLC lineage as shown by co-staining with Hand-r > LHG/LexAOP > H2B::mCherry (red) in 13 h.p.f. and 15 h.p.f. embryos in lateral **b** and dorso-lateral **c** views, respectively. aTLC: Anterior TLCs, pTLC: posterior TLCs. **d-l** Double fluorescent WMISH for *Esrp* (red) and *Twist-like2* (green), with the exception of **j**, which corresponds to colorimetric *Esrp* mRNA staining (purple). At middle tailbud stage, *Esrp* expression is detected in both epidermis and mesenchymal cell lineage (the latter is marked by arrows). **m-m'** Mesenchymal lineage in WT larvae at 18 h.p.f. stained in red using the Twist > CD4::mCherry construct. Twist > GFP was used as co-electroporation control plasmid, showing full co-electroporation (green channel not shown for clarity). **n-n'** Mesenchymal lineage stained in larvae co-electroporated with Twist > CD4::mCherry and Twist > Esrp constructs. The trunk region is shown in **m-n**, while tail segments are shown in **m'-n'**. **o** Quantification of the different phenotypes of mesenchymal cell lineage motility observed in Twist > Esrp and control larvae. These included individuals with abnormal migration in the trunk ('Trunk'), with ectopic mChe-positive cells in the tail ('Tail') or both phenotypes ('Trunk + Tail'). *P*-values correspond to a two-sided Fisher Exact tests ('Normal' vs rest). Scale bars correspond to 25 μm

during development (Fig. 1c). Although *esrp2* mutant females were unable to produce eggs, both mutant lines grew normally in homozygosity and became seemingly healthy adults. However, double mutants (DMUT) presented multiple developmental abnormalities. Most larvae (28/37, 75.7%) died between 8 and 10 d.p.f., and no fish survived beyond 14 d.p.f.. Phenotypic analysis of 6 d.p.f. DMUT larvae showed multiple fully penetrant morphogenetic defects. All DMUT larvae presented cleft palate (Fig. 1d), with the medial population of cartilage cells being absent in the ethmoid plate, as revealed by alcian blue staining (Fig. 1e). Fin development was also impaired, with evident dysgenesis of its distal endoskeletal part (Fig. 1f). Reduction in esophagus diameter and loss of its villous shape was observed (Fig. 1g). The volume of the inner ear was smaller compared to WT larvae, and the dorso-lateral septum that separates the rostral and caudal semicircular canals was abnormally invaded by cellular and extracellular material (Fig. 1h). The posterior part of the olfactory epithelium formed a spherical internal lumen, instead of being open toward the embryo surface (Fig. 1i). Mutants also showed an abnormal arrangement of basibranchial pharyngeal cartilage (Supplementary Fig. 3p), in addition to a smaller and thicker swim bladder epithelium (Supplementary Fig. 3q), which failed to inflate in 24/35 (68.5%) of examined DMUT embryos (Fig. 1c). Interestingly, *Esrp* genes in mouse are

also required during the development of structures homologous to some of these organs[15], including several that exhibit distinct morphologies and functions compared to zebrafish, such as the lungs/swim bladder, palate skeleton and pectoral limbs/fins.

To characterize the phenotype of the DMUT embryos at the molecular level, we performed RNA-seq of two replicates of DMUT 5 d.p.f. and age-matched WT larvae (Methods section). This analysis confirmed the reduced expression for both mutant alleles in DMUT embryos (Supplementary Fig. 3r). Differential gene expression analysis identified 248 and 609 downregulated and upregulated genes, respectively, (Supplementary Data 1). Gene Ontology (GO) enrichment analysis showed functional categories that were highly consistent with some of the observed phenotypes, such as skeletal system development and sensory perception (Supplementary Fig. 4). Interestingly, other significantly enriched categories pointed to specific impaired processes at the cellular level such as cell–cell adhesion, cell-matrix adhesion and cell component morphogenesis.

**Esrp expression and function during Ciona development.** To examine the diversity of roles that *Esrp* genes play during embryogenesis beyond the vertebrate clade, we next studied the ascidian *Ciona robusta*, a species belonging to the sister group of

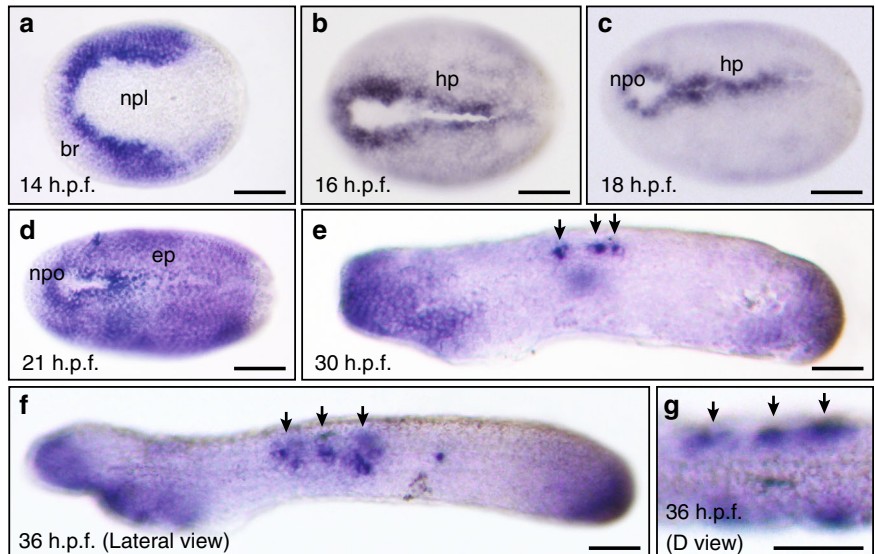

**Fig. 3** *Esrp* is expressed dynamically in the non-neural ectoderm during amphioxus embryo development. WMISH of *Esrp* in *Branchiostoma lanceolatum* embryos. Anterior is to the left in all cases. **a** 14 h.p.f. early neurula (dorsal view) showing expression in the ectodermal cells located in the border region (br) next to the neural plate (npl). **b**, **c** 16 h.p.f. and 18 h.p.f. mid-neurula embryos (dorsal view) stained most strongly in the ectodermal cells next to the neural plate border and that form the hinge points (hp) during neural tube closure. The location of the neuropore (npo) is indicated. **d** In 21 h.p.f. late neurula (dorsal view), *Esrp* expression is extended throughout the whole epidermis (ep). **e**, **f** Early (30 h.p.f.) and late (36 h.p.f.) pre-mouth stages (lateral views) showing *Esrp*-positive cells in anterior ectoderm, tailbud epithelia and in a few cells that likely corresponding to migrating sensory cells during epidermal incorporation or already integrated into the epithelium (black arrows). **g** Dorsal view of a 36 h.p.f. embryo shows the epidermal location those *Esrp*-positive cells (black arrows). Scale bars: 50 μm **a–f**, 25 μm **g**

vertebrates, the tunicates[17]. Expression of the single *Esrp* ortholog present in the *Ciona* genome was detected in the embryonic epidermis after neurulation, as in the case of vertebrates (Fig. 2). In addition, *Esrp* expression was also observed in a bilateral domain located within the mesenchymal lineage from early to late tailbud stage. *Ciona* mesenchymal cells derive from the *Twist-like1*-expressing A7.6 [trunk lateral cells; TLCs], B7.7, and B8.5 blastomeres of the 110-cell stage embryo[18]. As development proceeds, those cells divide and give rise to three mesenchymal sub-lineages that are located in a region of the trunk adjacent to the tail during mid-tailbud stages. *Twist-like1* enhances the transcription of several mesenchymal genes, including *Twist-like2*, before being downregulated around mid-tailbud stage[18,19]. In subsequent stages, a number of cells coming from the *Twist-like2*-positive lineage migrate toward the anterior part of the trunk to contribute to the formation of mesodermal organs[20].

We confirmed the expression of *Esrp* within the mesenchymal lineage with a double-staining assay. Fertilized eggs were electroporated with a construct (Twist > CD4::mCHe) carrying the *Twist-like1* promoter driving the expression of the membrane-targeted Cherry (mCHe) protein as a reporter[21]. An anti-mCHe antibody was used to track the mesenchymal cell lineage, while endogenous *Esrp* expression was visualized by fluorescent WMISH (Fig. 2a). A similar assay using a LexA/LexAop system driven by the *Hand-r* proximal enhancer[22] narrowed the identity of this *Esrp*-expressing domain down to a subset of posterior TLCs (Fig. 2b, c). This mesenchymal lineage contributes to the formation of specific organs like the oral siphon muscle and the epithelium of the 1st/2nd gill slits[20].

To unravel the dynamics of these *Twist-like1*-derived *Esrp*-positive cells, we next performed a double WMISH with probes for *Esrp* and *Twist-like2*. This showed that the expression of both genes is rapidly regulated during development and that they soon acquire a mutually exclusive expression pattern, which becomes evident by late tailbud stage (Fig. 2g–i). At late tailbud II stage,

*Twist-like2* was restricted to anterior mesenchymal domains, while *Esrp* transcripts were only detected in the posterior-most part of the trunk (Fig. 2j–l).

The common developmental origin of these two cell populations, the function of *Twist-like2* as a key inducer of cellular migration[23], and the described antagonistic roles of *Esrp* and *Twist* genes in mammalian cells[24] led us to hypothesize that *Esrp* may confer particular morphogenetic properties to the subset of cells within the mesenchymal lineage in which it is expressed and that, therefore, *Esrp* expression needs to be turned off in the other lineages for their correct ontogenesis. To test this possibility, we ectopically expressed *Esrp* in all mesenchymal cells (*Twist* > Esrp) during early stages, together with *Twist* > CD4::mCHe to visualize them. We fixed embryos at larval stage, when a large part of the cell migration towards the anterior part of the trunk has occurred. We observed two main phenotypes associated with an abnormal mesenchymal distribution in 75% of individuals (n = 70). In all affected embryos, migrating mCHe-positive cells were found only in the lateral sides adjacent to the epidermis, but not through the middle part of the trunk (Fig. 2m, n, o). In addition, 35% of co-electroporated larvae also showed ectopic mCHe-positive cells distributed along the tail, which were only very rarely observed in control individuals (Fig. 2o). Interestingly, a previous study found that some cells derived from B7.7 and B8.5 mesenchymal lineages, which normally do not express *Esrp*, localized in the tail of *Twist-like1* knockdown of *Ciona* larvae, integrating muscular and endodermal tissues[20]. Altogether, our results thus suggest that *Esrp* may modulate motility properties of the mesenchymal cell lineage in ascidians, as well as compromise topological cellular fate when ectopically expressed.

**Dynamic *Esrp* expression during amphioxus embryogenesis.** We next investigated *Esrp* expression in the amphioxus *Branchiostoma lanceolatum*, a cephalochordate species that shares a general chordate bodyplan with vertebrates, although it lacks

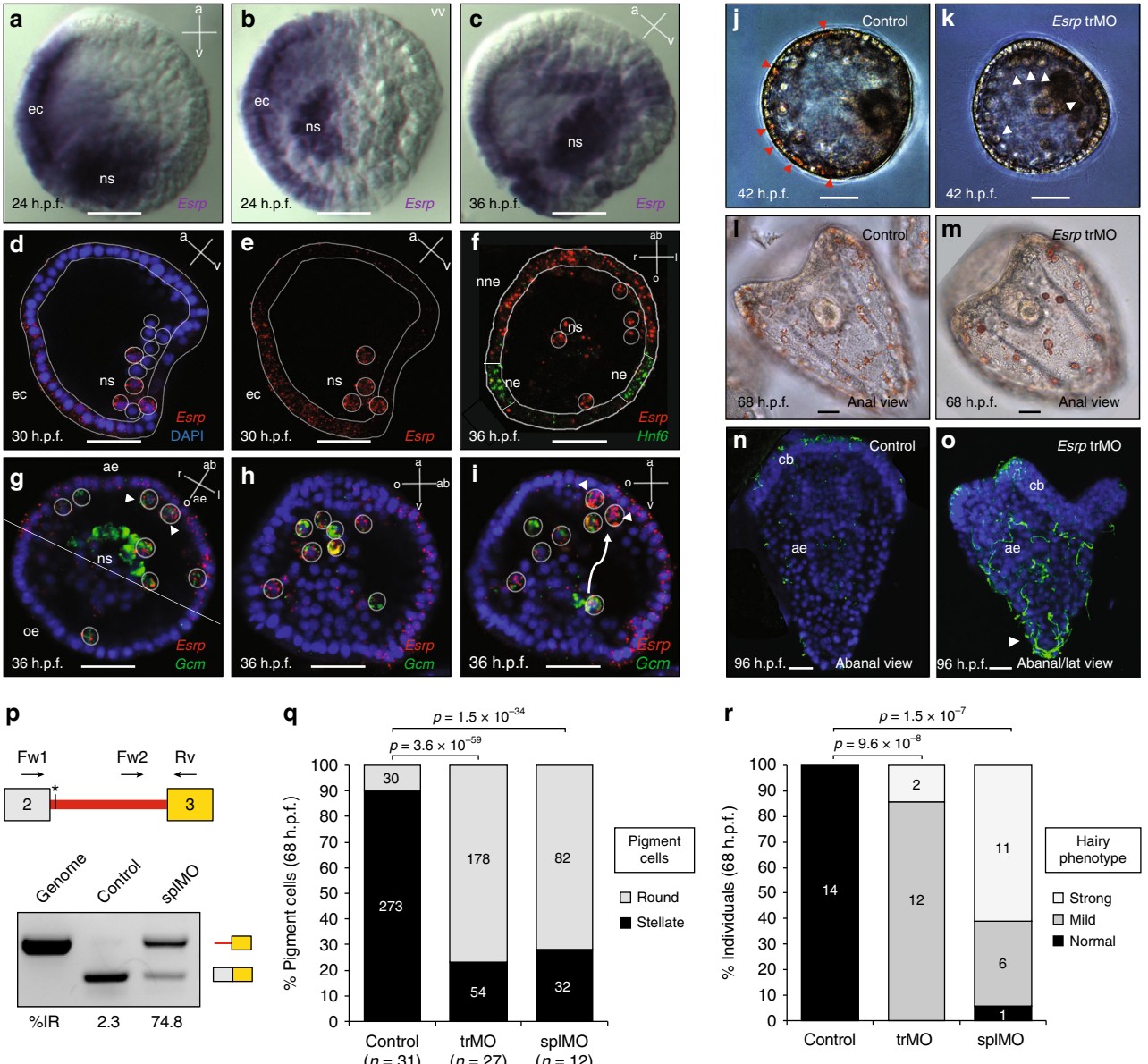

**Fig. 4** *Esrp* represses cilia formation in aboral ectoderm and is necessary for complete MET of pigment cells. **a**, **b** Colorimetric WMISH of *Esrp* in sea urchin embryos at 24 h.p.f., in lateral **a** and vegetal **b** views, showed expression in one side of the ectodermal territory (ec) and in some cells of the non-skeletogenic mesoderm (ns). vv, vegetal view. **c** Lateral view at 36 h.p.f. confirmed *Esrp*'s asymmetric expression in one side of the ns mesoderm. **d**, **e** Fluorescent WMISH at 30 h.p.f. (lateral view) confirmed ectodermal and mesodermal expression of *Esrp*. **f** Double fluorescent WMISH showed that *Esrp* expression (red) at 36 h.p.f. (animal pole view) did not overlap with the neurogenic ectoderm (ne) marker *Hnf6* (green). nne, non-neural ectoderm. **g** Double fluorescent WMISH of *Esrp* (red) and *Gcm* (green) at 36 h.p.f. (animal pole view) revealed that *Esrp* was expressed in the aboral ectoderm (ae) and in the pigment cell precursors. Pigment cell precursors that are already in contact with the ectodermal epithelium are marked by arrowheads. oe, oral ectoderm. **h**, **i** *Esrp* (green) and *Gcm* (red) at 36 h.p.f. in two different stacks from same embryo (lateral view). The arrow indicates a representative migratory path of pigment cells from mesoderm to ectoderm. At the top right corner of each panel **a**–**i**, the orientation of the depicted embryo along the animal [a] - vegetal [v], oral [o] – aboral [ab], left [l] –right [r] axes are reported, when possible. **j**, **k** Early gastrula (42 h.p.f.) from uninjected control and *Esrp* trMO embryos showed differences in pigment cell location. Red/white arrowheads indicate pigment cells already integrated into the ectoderm or in the sub-ectodermal space, respectively. **l**, **m** Early pluteus larvae (68 h.p.f.) in abanal view show differences in pigment cell morphology upon trMO treatment (roundish instead of stellate/dendritic). **n**, **o** Ectopic embryonic cilia stained with acTubulin in the aboral ectoderm (ae) of 96 h.p.f. *Esrp*-trMO injected embryos. Apex is marked by a white arrowhead. cb, ciliary band. **p** RT-PCRs showing the levels of intron 2 retention in *Esrp* transcripts from control and *Esrp* splMO embryos; genomic DNA was used as a reference for intron inclusion. The asterisk marks the position of the first in-frame termination codon in intron-retained transcripts. **q** Quantification of pigment cell morphology in control, *Esrp* trMO and splMO knockdown 68 h.p.f. embryos (sum of three independent experiments). **r** Quantification of the 'hairy' phenotype in control, *Esrp* trMO and spMO knockdown 68 h.p.f. embryos (sum of two independent experiments). *P*-values correspond to 2-way **q** or 3-way **r** two-sided Fisher Exact tests. Hairy phenotype was considered "mild" when aboral ectoderm cells show long cilia (of similar or longer length than ciliary band cells) and "strong" when, in addition to the latter, larvae show particularly long cilia at the apex, as indicated by an arrowhead in panel **o**. Scale bars correspond to 20 μm

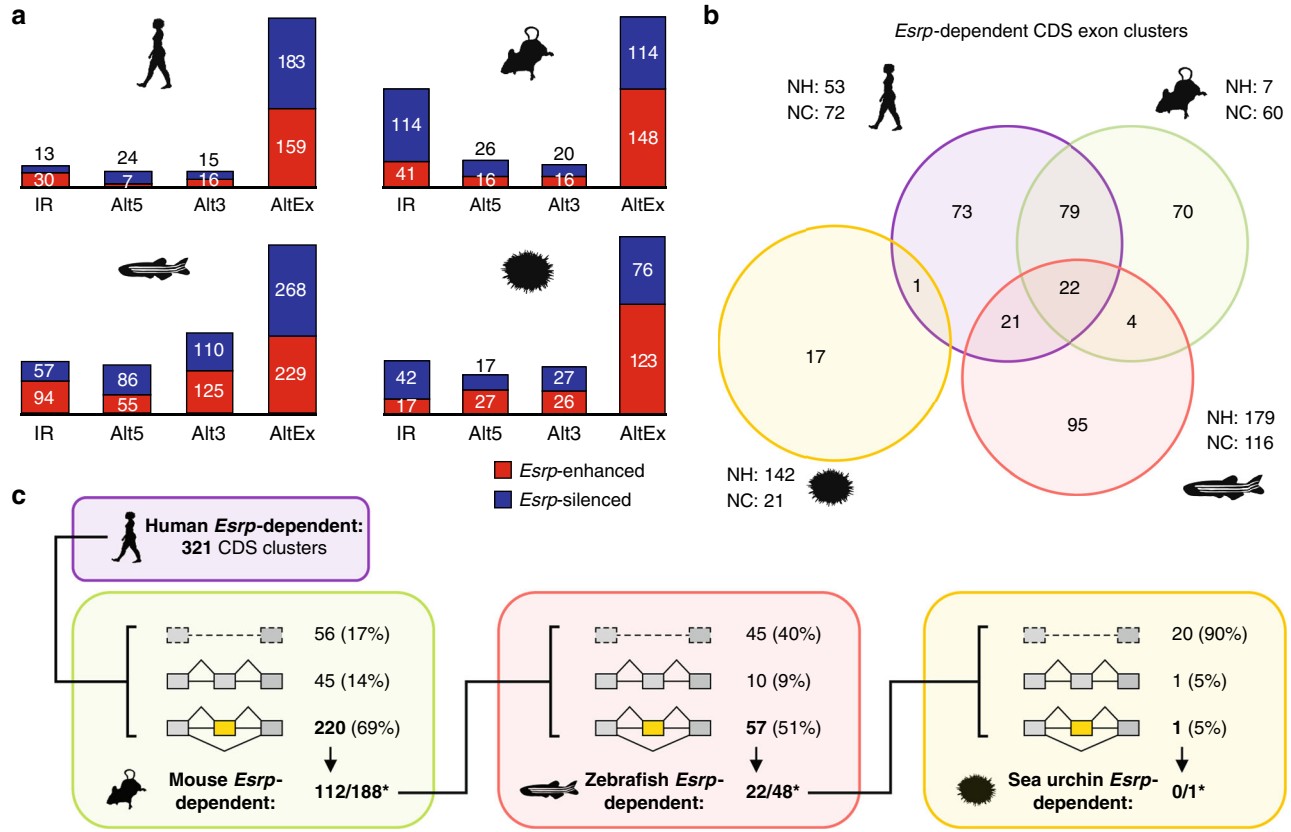

**Fig. 5** Evolution of *Esrp*-dependent splicing programs. **a** Number *Esrp*-dependent AS events by type detected in human, mouse, zebrafish and sea urchin, RNA-seq samples. Red/blue bars correspond to *Esrp*-enhanced/silenced inclusion of the alternative sequence. IR, intron retention; Alt5, alternative 5′ splice site choice; Alt3, alternative 3′ splice site choice; AltEx, alternative cassette exons. **b** Venn diagram showing the overlap among homologous *Esrp*-dependent cassette exons in coding regions detected as regulated in the same direction in the studied species. Only *Esrp*-dependent exons with homologs in at least two species and sufficient read coverage in all the species in which they have homologs are included in the comparison. *Esrp*-dependent exons that lack homologous counterparts in all the other species are indicated for each species (NH). Similarly, the number of *Esrp*-dependent exons that do not have sufficient read coverage in at least one of the species with a homologous exon is displayed for each species (NC). **c** Summary of conservation at the level of genomic presence, alternative splicing and *Esrp*-dependency for all 321 clusters of human *Esrp*-dependent coding exons in other species. Shared *Esrp*-dependent exons between the previous phylogenetic group are classified in the test species into three categories: (i) the exon is not detected in the genome (top row, discontinuous line); (ii) the homologous exon is detected in the genome, but it is constitutively spliced (middle row, gray exon); and (iii) the homologous exon is alternatively spliced (bottom row, yellow exon, numbers in bold). Below this classification, the number of similarly regulated *Esrp*-dependent exons shared by the two species/lineages over the number of alternatively spliced exons with sufficient read coverage from (iii) (indicated by asterisks), is shown

many of their key traits and presents several developmental particularities[25]. Amphioxus *Esrp* showed a highly dynamic expression pattern during embryo development, generally more restricted than its vertebrate counterparts (Fig. 3). In early neurula embryos (14 h.p.f.), *Esrp* transcripts were observed in the border region, an ectodermal tissue adjacent to the neural plate (Fig. 3a). During neurulation, *Esrp* was strongly expressed along the ectodermal hinge points of neural tube closure (Fig. 3b, c). Right after neurulation (21 h.p.f.), transcription of *Esrp* was extended to the epidermis of the whole embryo (Fig. 3d). Finally, in pre-mouth larvae stages, the expression was restricted to the tailbud region, anterior ectoderm, and, strikingly, in a few cells near and within the dorsal epidermis (arrows in Fig. 3e–g). Because of their shape and location, the latter group of cells may correspond to a previously described population of epidermal sensory neurons[26–28]. These cells have been reported to delaminate from the ventral ectoderm, migrate underneath the epithelium toward the dorsal part of the embryo and re-integrate into the ectoderm, where they become sensory cells. Based on its dorsal expression, we speculate that *Esrp* might contribute to the process of epithelial integration at the final migratory stages.

**Esrp is required for MET of pigment cells in sea urchin**. We next investigated the functions of *Esrp* in the purple sea urchin *Strongylocentrotus purpuratus*. This organism belongs to Ambulacraria, sister group of chordates, with a transcriptome-based annotated genome and available genetic tools. Colorimetric WMISHs revealed *Esrp* expression in the ectoderm and mesoderm in one half of the embryo at blastula stages (24–36 h.p.f.) (Fig. 4a–c). Expression was no longer detected at gastrula (48 h.p.f.) or early pluteus stages (72 h.p.f.). Fluorescent WMISH confirmed *Esrp* expression in the ectoderm and non-skeletogenic mesodermal cells at 30 h.p.f. (Fig. 4d, e). To investigate the nature of the ectodermal region with *Esrp* expression, we performed double fluorescent WMISH with *Esrp* and *Hnf6*, a marker for the ciliary band[29], the region where ciliary cells and neurons differentiate (Fig. 4f). No signal overlap was observed between the two genes, indicating that *Esrp* expression is restricted to the non-neural ectoderm. Next, double WMISH with *Gcm*, a marker for aboral non-skeletogenic mesoderm at the mesenchyme from blastula stage onward[30], showed that *Esrp* expression in the ectodermal tissue corresponds to the aboral side (Fig. 4g–i). Interestingly, we observed co-expression of *Gcm* and *Esrp* in pigment cells precursors, a population of non-skeletogenic

mesodermal cells that delaminate from the aboral side of the tip of the invaginating archenteron and migrate during gastrulation from the mesoderm toward the ectoderm, where they insert between epithelial cells becoming immunocytes[31,32].

To explore the functions of *Esrp* in this organism, we generated gene knockdowns by injecting morpholinos (MO) into sea urchin zygotes. To ensure that the knockdowns produced specific effects, we used two different MOs in independent experiments: one blocking translation and another impairing splicing of intron 2, whose retention generates a premature termination codon four aminoacids downstream of exon 2. Efficiency of the splicing MO was assessed by RT-PCR (Fig. 4p). We observed two main phenotypic defects in both MO injections, supporting knockdown specificity (Fig. 4q, r). First, whereas in control embryos most pigment cells were already located in the ectoderm at gastrula stage (42 h.p.f.) (Fig. 4j, k, Supplementary Fig. 5a) and acquired a dendritic conformation by prism/early pluteus stages (68 h.p.f.) (Fig. 4l, m, q), in knockdown embryos these cells were usually observed in the sub-ectodermal space at gastrula stage and maintained their roundish shape at prism stages ($p < 10^{-4}$ for all comparisons, Fisher Exact test). This failure in the complete integration of pigment cells into the ectoderm likely constitutes an impaired MET. Second, knockdown embryos showed a "hairy" phenotype at gastrula and pluteus stages, consisting of ectopic long cilia on the aboral ectoderm, especially at the apex (Fig. 4j, k, n, o, r, Supplementary Fig 4b; $p < 10^{-6}$ for all comparisons, 3-way Fisher Exact test). In summary, these results indicate that *Esrp* in sea urchin is necessary for a complete integration of pigment cells into its destination epithelium and to avoid ciliogenesis in aboral ectodermal cells.

To further characterize these developmental defects, we generated RNA-seq data for two replicates of SplMO-injected embryos at 24 h.p.f. and age-matched controls. These data confirmed our RT-PCRs results showing a high level of retention of *Esrp* intron 2 due to the morpholino effect (Supplementary Fig. 5c). Differential gene expression analysis identified 360 and 712 downregulated and upregulated genes, respectively (Supplementary Data 2). Interestingly, some enriched GO categories for these gene sets were similar to those found for the zebrafish DMUT analysis, including cell–cell adhesion, cell component morphogenesis and ectoderm development (Supplementary Fig. 5d). We also observed a significant enrichment for functions related to nervous system development among the upregulated genes, consistent with a possible failure in proper non-neural ectoderm specification.

**Evolutionary comparison of *Esrp*-dependent splicing programs.** Next, we investigated the origin and evolution of alternative exon programs regulated by *Esrp* across multiple phylogenetic distances using our zebrafish and sea urchin RNA-seq data, as well as previously published RNA-seq data for *Esrp* perturbations in mouse epidermis[15] and three human cell cultures[33,34]. Importantly, this approach detects both direct and indirect *Esrp*-regulated events, hereafter referred together as *Esrp*-dependent. We identified 342 differentially spliced cassette exons in human, 262 in mouse, 497 in zebrafish, and 199 in sea urchin (Supplementary File 1; see Methods section for details). In all species, these exons were found enriched in genes associated with certain GO terms, such as vesicle-mediated transport, GEF activity, and actin cytoskeleton cell components (Supplementary Fig. 6). Additionally, we detected lower numbers of other types of AS regulatory changes, such as alternative 5′ or 3′ splice site choice and intron retention (Fig. 5a, Supplementary Data 3). RT-PCR assays validated all tested differentially regulated exons in zebrafish (ΔPSI correlation $R^2 = 0.90$, $n = 15$) and in sea urchin

($R^2 = 0.89$, $n = 12$) (Supplementary Fig. 7). RNA regulatory maps for the top twelve hexamers bound by *Esrp* identified by SELEX-seq[33] showed clear enrichment above background at expected regions for *Esrp*-dependent exons for all species (Supplementary Fig. 8a), although results from sea urchin were less clear, likely reflecting a higher fraction of indirect targets detected using a knockdown strategy in whole embryos.

To understand the evolution of *Esrp*-dependent programs at the exon level, we next built clusters of high confidence homologous coding cassette exons within conserved gene intron-exon structures (Supplementary Data 4 and see Methods section). Comparisons between closely and distantly related organisms uncovered different scenarios with regards to the sources for exon recruitment. On the one hand, most *Esrp*-dependent exons among vertebrates presented a homologous counterpart in another species, ranging from 97% in mouse to 59% in zebrafish (Supplementary Fig. 8b). On the other hand, we only detected homologous counterparts in any of the studied genomes for 21% of sea urchin differentially regulated exons. When taking only the clusters with homologous exons in at least two species and with sufficient read coverage in all the organisms in which the exon exists, both mammals shared most of their *Esrp*-dependent exon sets (Fig. 5b, Supplementary Fig. 8c). On the contrary, we observed a large fraction of lineage-specific *Esrp*-dependent regulation in the case of zebrafish (67%) and sea urchin (94%). Alternative exons with *Esrp*-like motifs in expected exonic or nearby intronic regions in vertebrates generally showed higher levels of shared *Esrp*-dependent regulation (Supplementary Fig. 9a), and presence of *Esrp*-like motifs in equivalent positions in orthologous exons was often associated with increased shared regulation (Supplementary Fig. 9b). Consistently, human exons with shared *Esrp*-dependent regulation between mammals or among vertebrates showed higher conservation of their flanking intronic sequences than human-specific and non-*Esrp*-dependent alternative exons (Supplementary Fig. 8d).

When focusing on the full set of 321 clusters with human *Esrp*-dependent exons, 83% had a homologous exon in mouse, most of which were also alternatively spliced in the rodent species (Fig. 5c). Among the homologous alternative exons with sufficient read coverage in mouse, 112/188 (60%) were identified as *Esrp*-dependent (in the same direction) also in this species. These events include numerous previously described *Esrp*-dependent exons in genes, such as *Scrib*, *Nf2*, *Enah*, and *Grhl1*[15]. From this set of shared mammalian targets, we found homologous exons in zebrafish in 67 cases (60%), most of which were also alternatively spliced in this species (Fig. 5c). Interestingly, we detected a core set of 22 homologous exons classified as *Esrp*-dependent in the three vertebrate species, including some in genes previously associated with morphogenetic processes[35] (e.g., *Numb*, *p120-catenin*, *Arhgap17* or *Itga6*; Supplementary Table 1). However, most of these exons (90%) did not have a detected homologous counterpart in sea urchin, and the two exons identified in the echinoderm genome did not exhibit *Esrp*-dependent differential regulation (Fig. 5c).

We also identified 49 orthologous genes whose AS was dependent on *Esrp* in at least two species, but in which the specific regulated exons were different (i.e., non-homologous). Remarkably, 21 of these cases involved sea urchin and at least one vertebrate. Moreover, we further observed a significant fraction of target genes with more than one *Esrp*-dependent exon, ranging from 5.2% in sea urchin to 19.1% in zebrafish. These observations highlight the evolutionary plasticity of some genes for multiple acquisition of *Esrp*-dependent AS regulation. An interesting case is *Cd44*, which acquired at least nine *Esrp*-dependent novel exons within the mammalian clade, and is involved in ureteric branching in mouse[36], a process affected in *Esrp1* KO mice[37].

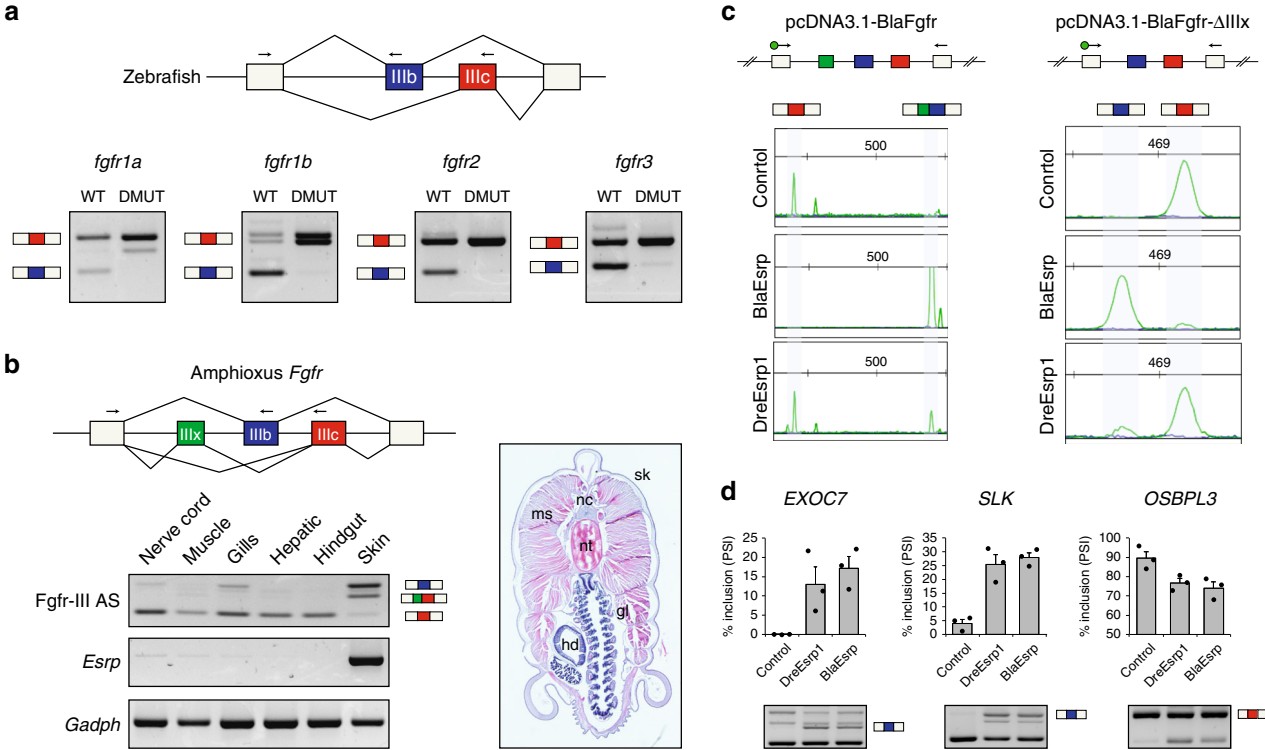

**Fig. 6** *Fgfr* AS is regulated by *Esrp* genes in vertebrates and amphioxus. **a** RT-PCR assays showing differential *Fgfr* exon IIIb and IIIc inclusion in WT versus DMUT 5 d.p.f. zebrafish embryos. **b** RT-PCR assays for *Fgfr* AS in different amphioxus adult tissues, depicted in a transversal section. nc, nerve cord, ms, muscle, gl, gills, hd, hepatic diverticulum, nt, notochord, sk, skin. Reverse primers were designed in both exons IIIb and IIIc (arrows) and used together in the same PCR reaction. **c** Top: schematic representation of pcDNA3.1-based minigene constructs containing the genomic region spanning the *Fgfr* AS event of *Branchiostoma lanceolatum*, with (pcDNA3.1-BlaFGFR) and without (pcDNA3.1-BlaFGFRΔIIIx) exon IIIx. Bottom: relative intensity of fluorescent RT-PCR bands supporting differential inclusion of exons IIIb and IIIc when transfecting the minigenes alone (Control) or together with a plasmid containing either amphioxus or zebrafish full-length *Esrp* transcripts (BlaEsrp and DreEsrp1, respectively). Despite significant mis-splicing of the minigene in all conditions, only the amphioxus construct was able to induce a dramatic switch toward exon IIIb inclusion. Primers were designed in the neighboring constitutive exons (arrows). **d** RT-PCR assays for endogenous human AS events in the same control, BlaEsrp or DreEsrp1 transfected 293T cells showing that the amphioxus and zebrafish *Esrp* constructs are able to modulate endogenous *Esrp*-dependent events in a similar manner. Error bars correspond to standard errors of three biological replicates. *Esrp*-enhanced isoforms are marked with an isoform cartoon

Finally, it should be noted that additional transcriptomic data from other tissues or developmental stages may increase the fraction of shared *Esrp*-dependent exons detected among species, whereas the use of techniques such as CLIP-Seq would allow actual discrimination between direct and indirect targets in future studies.

**Regulation of *Fgfr* AS by Esrp is conserved among chordates.** We next focused on the AS event in the *Fgfr* family, which is the only one of the 22 *Esrp*-dependent homologous exon groups shared by the three vertebrate species that affects multiple paralogs in each organism. Evolutionary conservation of the mutually exclusive exons encoding the IgIII domain of *fgfr2* had been previously reported for zebrafish[38]. As in mammals[39], we found homologous AS events also in *fgfr1a*, *fgfr1b* and *fgfr3*, but not in *fgfr4*. RT-PCR assays for those *Fgfr* genes showed a complete isoform switch toward the mesenchymal IIIc exons in zebrafish *esrp1* and *esrp2* DMUT embryos (Fig. 6a). Interestingly, the *Fgfr* ortholog of sea urchin harbored only one exon homologous to those alternatively spliced in vertebrates, which was constitutively included in all transcripts (see below). Therefore, to assess when the vertebrate *Fgfr* AS event originated during evolution, we then turned to the chordate amphioxus. We found that the sole amphioxus *Fgfr* gene harbors exons homologous to IIIb and IIIc that are also alternatively spliced in a mutually exclusive manner.

In addition, a cephalochordate-specific exon was found in this genomic region (exon IIIx; Fig. 6b). RT-PCR assays on dissected adult tissues showed that *Esrp* expression was only strongly detected in the amphioxus skin, where the highest inclusion of exon IIIb was also observed (Fig. 6b). An amphioxus-specific isoform including both exons IIIx and IIIc was also detected in this tissue, while inclusion of exon IIIc alone was nearly absent. On the other hand, the rest of the tissues express predominantly the isoform containing only the exon IIIc.

To gain further insights into the regulatory evolution of this event, we made two minigene constructs, one comprising the whole genomic region spanning the amphioxus *Fgfr* AS event, and one lacking the amphioxus-specific exon IIIx (Fig. 6c). These minigenes were individually transfected into human 293T cells together with an expression vector containing the amphioxus or the zebrafish full-length *Esrp* transcripts. Remarkably, despite considerable mis-splicing of the minigenes in 293T cells, amphioxus *Esrp* acted as a major regulator of *Fgfr* AS, promoting inclusion of IIIb exon in both minigenes (Fig. 6c). On the other hand, zebrafish *esrp1* produced only very mild changes compared to the control, despite the fact that both amphioxus and zebrafish *Esrp* genes were able to modify the AS pattern of endogenous exon targets as the human *ESRP1* (Fig. 6d, Supplementary Fig. 10). Therefore, altogether, these results provide two main insights into the evolution of *Fgfr* AS regulation: (i) its mutually exclusive regulation by *Esrp* originated before the last common

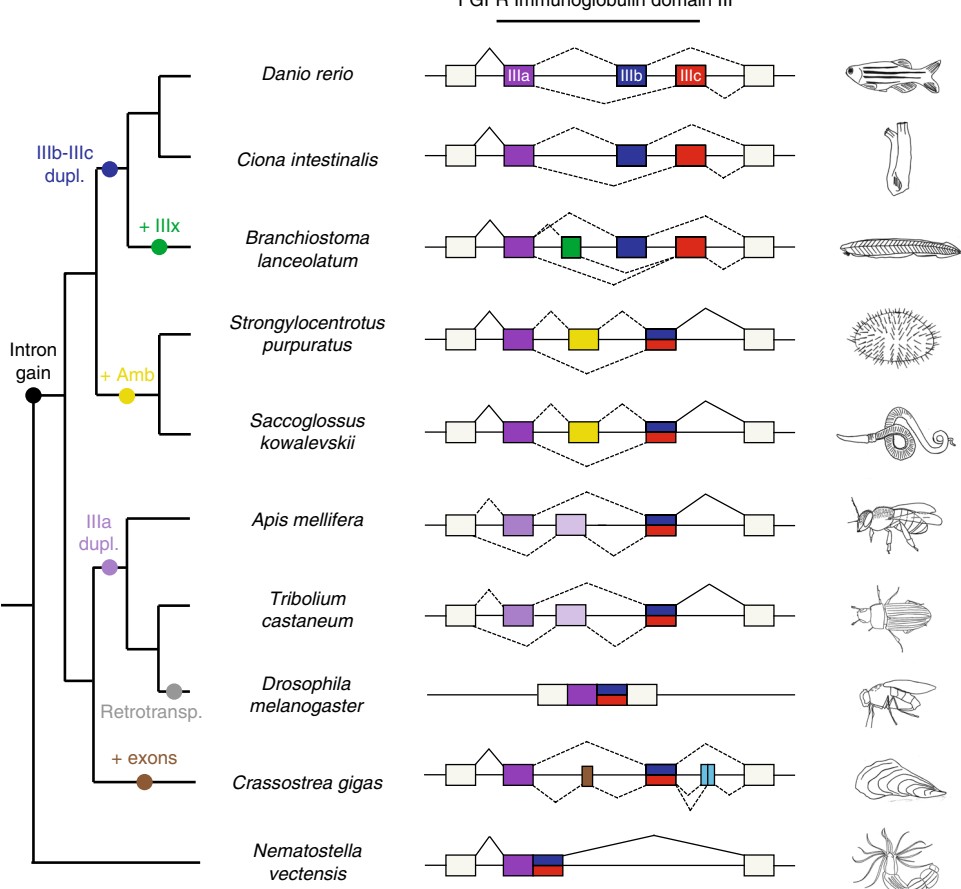

**Fig. 7** Gene structure and AS at the IgIII domain of the *Fgfr* gene family in metazoans. Schematic representation of the AS diversity in the region encoding the homolog of the *Fgfr* IgIII domain for zebrafish (*Danio rerio*), vase tunicate (*Ciona intestinalis*), amphioxus (*Branchiostoma lanceolatum*), purple sea urchin (*Strongylocentrotus purpuratus*), acorn worm (*Saccoglossus kowalevskii*), honey bee (*Apis mellifera*), red flour beetle (*Tribolium castaneum*), fruit fly (*Drosophila melanogaster*), pacific oyster (*Crassostrea gigas*), and starlet sea anemone (*Nematostella vectensis*). Boxes represent exons, horizontal lines are introns, and diagonal lines connect splicing junctions. Homologous to vertebrate exons IIIa, IIIb, and IIIc are shown in purple, blue and red, respectively. The non-chordate pro-orthologous exon of exons IIIb and IIIc is colored half blue and half red. The amphioxus- and ambulacrarian-specific alternative exons are depicted in green and yellow, respectively. Light violet colors are used for the insect-specific mutually exclusive event involving exon IIIa, and brown and light blue for oyster-specific exons. Gray exons are constitutive in all species. Fruit fly orthologs (*heartless* and *branchless*) are intronless genes. Dupl: duplication; Amb: ambulacraria-specific exon; retrotransp.: retrotranscription

ancestor of chordates; and (ii) it exhibits lineage-specific regulatory requirements in *trans* in cephalochordates.

**Independent evolution of AS in *Fgfr* in Bilaterian lineages**. Given its remarkable conservation across chordates for over 550 million years of independent evolution, we next studied the evolution of *Fgfr* AS in further detail. A gene structure analysis for Bilateria species showed that vertebrate exons IIIb and IIIc are the result of a single tandem exon duplication that occurred in the lineage to leading to chordates, before the split of its three main sub-phyla (Fig. 7). In contrast, the single pro-orthologous exon IIIb/IIIc is found as constitutively included in all observed transcripts from all studied non-chordate species. However, previous studies reported other AS events in the genomic locus encoding the region homologous to the IgIII domain in two non-vertebrate species (the purple sea urchin[40,41] and the red flour beetle[42]), and we further show that a diverse array of AS events independently evolved in most studied lineages (Fig. 7). In particular, the previously reported alternative exon in sea urchin is located upstream of exon IIIb/IIIc and exhibits no sequence similarity with its neighboring exons. Moreover, its inclusion was not detected as *Esrp*-dependent, which is consistent with the non-

overlapping expression of *Fgfr* and *Esrp* genes during sea urchin's development[43]. Remarkably, this exon is also present an alternatively spliced across the Ambulacraria clade, which comprises echinoderms and hemichordates. In summary, our phylogenetic survey shows that this genomic region is a hotspot for recurrent AS evolution across Bilateria, likely due to its potential to modulate FGF signaling[8].

## Discussion
Similar to recent reports in mouse[13,15], we found that *Esrp* genes are dynamically expressed in non-neural ectoderm territories during some developmental stages of zebrafish, vase tunicate, amphioxus, and sea urchin. This suggests a putative ancestral regulatory role for *Esrp* in embryonic non-neural ectoderm at least since the last common ancestor of living deuterostomes. Additionally, *Esrp* was also detected in these organisms in many other tissues involved in clade-specific morphogenetic processes. While this gene family is likely to play a wide variety of roles depending on the cell type and species during development, in some cases *Esrp* seems to influence epithelial integration and/or cell motility properties. For instance, in sea urchin, mesodermal *Esrp*-expressing cells undergo a MET, thereby becoming

integrated into ectodermal epithelium, a process that was impaired upon *Esrp* knockdown. Moreover, a similar function may explain *Esrp* expression in a group of amphioxus migrating cells. In the vase vase tunicate, our experiments suggest that co-option of *Esrp* into a subpopulation of the mesenchymal cell lineage may have contributed to modulate their motility. And, in vertebrates, a population of neural crest-derived migratory cells[44] that integrate into the palate are absent in this structure in both mouse and zebrafish DMUT embryos. Moreover, in human cell cultures and cancer, *ESRP1* has been shown to increase cell adhesion and reduce cell motility[24,35]. Altogether, these results suggest that, although they are usually involved in other types of ontogenetic functions in vertebrates, like branching morphogenesis[15,37,45], *Esrp* might also contribute to epithelial integration and/or cell motility during some specific developmental processes in this subphylum. Thus, given this phylogenetic distribution, it is plausible that *Esrp* may have regulated specific MET processes in the ancestor of living deuterostomes.

Despite these suggested ancestral roles, many of the detected exons showed clade-restricted regulation. Such diversification of transcriptomic targets may have allowed the refinement and expansion of *Esrp* ontogenetic functions in each lineage. Interestingly, among studied vertebrates, non-overlapping programs have evolved mostly by recruiting pre-existing exons, which were usually already alternatively spliced. On the other hand, only a minor fraction of the echinoderm *Esrp*-dependent exons have detected homologous counterparts in any of the vertebrate genomes (and vice versa). This indicates that origin of novel exons is a major factor contributing to re-assembly of lineage-specific *Esrp* regulatory programs among distantly related metazoan lineages, as it has been suggested for other AS factors[46–48]. However, we identified several gene functions that were enriched among *Esrp*-dependent programs across the studied species, altogether indicating that *Esrp* regulation may be impacting some common molecular pathways. Moreover, we observed several cases of shared *Esrp* regulation at the gene level in which the specific exons differ in each phyla, including genes involved in diverse morphogenetic processes in metazoans[49–53] (e.g., *Exoc7*, *Slain2*, *Epb41*, *Ift88*, and *Scrib*). These could provide an explanation for the progressive disentanglement of exon programs across phylogeny while still having an impact on some related molecular functions.

Our results also highlight multiple conserved roles of *Esrp* in specific organogenetic processes within bony vertebrates, as similar phenotypes in various homologous structures are found in both mouse and zebrafish mutants. This is reflected at the molecular level by a core target set of at least 22 homologous exons among the three studied vertebrate species, including members of the *Fgfr* family. In fact, a number of impaired structures reported in esrp1 and esrp2 DMUT embryos have also been described in conditional mutant mice for the *Fgfr2*-IIIb epithelial isoform[10]. This is consistent with our results that zebrafish *Esrp* genes, as well as their mammalian counterparts, are essential for inclusion of IIIb exons in embryonic tissues. Thus, a major molecular cause for the phenotypes reported in mouse and zebrafish DMUTs is likely the disturbance of epithelial-mesenchymal FGF signaling during development of those organs. This case illustrates how the recruitment of one or few key targets can both constrain and canalize the function of a regulatory factor within a certain clade.

Finally, we determined that *Esrp*-dependent AS of *Fgfr* genes originated in stem chordates, after intragenic tandem duplication of the pro-homologous exon IIIb/IIIc. Thereby, chordates evolved a unique way of using the FGF signaling to regulate epithelial-mesenchymal crosstalk during development through a new molecular interaction that has been intensively exploited by vertebrates. Strikingly, different AS events have independently evolved in other phyla in the orthologous genomic region of *Fgfr* genes encoding the IgIII domain. We suggest that most of these AS events in non-chordate species may also contribute to modulate the affinity of the receptors for different *Fgf* ligands, although their specific developmental roles will be determined by the splicing regulatory logic in each organism. Furthermore, this case exemplifies how a probably non-adaptive fixation at the microevolutionary level had a long-term impact on animal evolution: an intron gain mutation provided the Bilateria lineage with a hotspot of great macroevolutionary potential through post-transcriptional regulation.

## Methods

**Domain and phylogenetic analyzes of *Esrp* genes.** Putative *Esrp* orthologs for all organisms included in Supplementary Figs. 1, 2 were identified by combining blastp and tblastn searches against NCBI databases and resources specific for sponges[54] and choanoflagellates (unpublished transcripts provided by Daniel Richter). Domain detection was performed using the NCBI conserved domain search function[55] (https://www.ncbi.nlm.nih.gov/Structure/cdd/wrpsb.cgi). To reconstruct the phylogenetic relationships of *Esrp* genes among the Apoikozoa clade (Metazoa + Choanoflagellata), we first aligned protein sequences with MAFFT[56]. Neighbor-joining algorithm, as implemented by MEGA7 with default parameters and a JTT substitution model, was employed for tree reconstruction using only conserved positions of the multi-sequence alignment, as identified by MAFFT.

**Zebrafish experimental procedures.** Breeding zebrafish (*Danio rerio*) were maintained at 28 °C on a 14 h light/10 h dark cycle as previously described. All protocols used have been approved by the Institutional Animal Care and Use Ethic Committee (PRBB–IACUEC), and implemented according to national and European regulations. All experiments were carried out in accordance with the principles of the 3Rs.

To investigate the expression of *Esrp* genes during development, embryos were raised at 28 °C for staging[57] and fixed overnight with 4% paraformaldehyde (PFA) in PBS at 4 °C. RNA probes were labeled with digoxigenin, and WMISH was performed using Nitrobluetetrazolium/bromochloroindolyl phosphate (NBT/BCIP) as chromogenic substrate for the final alkaline phosphatase[58]. A minimum of 15 embryos of the same stage was used to evaluate expression patterns. To create loss-of-function zebrafish lines for esrp1 and esrp2, we used the CRISPR-Cas9 system. We first assessed the presence of multiple promoters in both genes looking at RNA-seq-based annotations and H3K4me3 ChIP-seq peaks at their respective loci in the UCSC browser. Since each gene had only one promoter, we designed single-guide RNAs (gRNAs) targeting the first and third exons of esrp1 and esrp2, respectively. The genomic target sites were identified using a publicly available web tool (http://crispr.mit.edu/). Selected targeted gRNA sequences corresponded to: 5′-GGAGCAAGTGGGGATAAGTTGGG-3′ for esrp1 and 5′-GGAGACCGGGCTCACTGCCGAGG-3′ for esrp2. The CRISPR-Cas9 approach was performed following the protocol from Chen and Wente laboratories[59]. Engineered vectors were obtained from Addgene. Volume of 1 nl of a mixed solution containing gRNA (80 ng/μl) and purified Cas9 mRNA (150 ng/μl) was microinjected into one-cell stage zebrafish embryos. F0 founders were crossed with a WT AB strain, and F1 individuals genotyped by fin clipping to select the appropriate mutations. We selected the following mutations for further analyzes: a 168-bp deletion together with a 14-bp insertion that induced the usage of a cryptic splice donor upstream the start codon for esrp1, and a 17-bp frame-disrupting deletion for esrp2 (Fig. 1b). Crosses between male and female heterozygous individuals carrying the same mutation were set to obtain single-mutant lines in the F2 generation. In addition, individuals from these esrp1 and esrp2 F1 lines were crossed to obtain a F2 generation of double heterozygous. DMUT embryos were obtained in subsequent F3 generations from intercrosses between F2 double heterozygous fish, at the expected Mendelian ratio. Expression of esrp1 and esrp2 in WT, single and double mutant embryos was assessed by quantitative PCR using Life Cycler 480 (Roche) with RNA extractions obtained from pools of 12 embryos. Relative expression values were obtained by normalizing against *Elfa* housekeeping gene[60]. Sequences of primers used for genotyping and quantitative PCR assays are provided in Supplementary Table 2.

For histological analysis, fixed embryos were progressively dehydrated in ethanol and xylol, embedded in paraffin, sectioned in transversal orientation with a microtome at 7–10 μm thickness, stained with haematoxilyn and eosin, and mounted with DPX (Eukitt). Alcian blue staining of chondrocytes at 6 d.p.f. embryos was performed according to the Zebrafish Book[61]. At least 18 WT and DMUT embryos were used for histological analyzes.

**Protein extraction and western blot assays.** Total cellular proteins were extracted from a pool of embryos (~35 animals) in 0.25 ml of RIPA buffer (10 mM Tris-HCl, pH 7.4; 1 mM EDTA; 150 mM, NaCl; 1% Triton X-100; 0.1% SDS;

protease inhibitor cocktail tablet [Roche]). After incubation for 20 min at 4 °C, homogenization was done using the Tissue-lyser-II (Qiagen). A centrifugation step was performed at 4 °C for 20 min at 14,000×g to remove cell debris. The supernatant containing the proteins was transferred into a new tube and stored at −80 °C until use. The Bio-Rad Protein Assay, based on the method of Bradford, and subsequent measurement at 595 nm with a spectrophotometer was used for determining protein concentration, and use of a standard curve with known amounts of BSA provided an estimate of the relative protein concentration.

For western blot assays, protein samples were heated at 70 °C for 10 min in presence of NuPage LDS Sample buffer 4× containing 1 μl NuPage reducing agent 10× (Invitrogen, Life Technologies). Total proteins were fractionated by SDS-PAGE on NuPage 4–12% Bis-Tris Gel (Invitrogen, Life Technologies) in MES buffer (2-[N-morpholino] ethanesulfonic acid) at a constant 200 V and transferred for 2 h at 200 mA to a 0,20 μm nitrocellulose membrane/filter paper sandwich included with the NuPAGE Large Protein Blotting Kit. The proteins were stained with PONCEAU-RED solution (Sigma-Aldrich) and after blocking with 5% non-fat milk in TBST (10 mM Tris, pH 8.0; 150 mM NaCl; 0.5% Tween-20) for 60 min, the membrane was incubated with primary antibodies against zebrafish ESRP1 1:800 (custom production by ChinaPeptides Co. against "NH2-QVMADHLNVAVDSSLHAFTAYK–C" epitope), mouse ESRP2 1:800 (NBP2-13972, Novus Biologicals) and zebrafish α-tubulin 1:5000 (T5168, Sigma-Aldrich) overnight at 4 °C. Membranes were washed three times with TBST for 10 min and incubated with a dilution of horseradish peroxidase-conjugated anti-mouse or anti-rabbit antibodies for 1 h at RT. Blots were washed with TBST three times and developed with the ECL system (Amersham) according to the manufacturer's protocols. Antibodies against ESRP1 and ESRP2 yielded strong bands of the expected size in the WT samples that fully disappear in the DMUT embryos. In addition, shorter weaker bands were observed for both antibodies which in most cases likely corresponded to unspecific signal (asterisks in Supplementary Fig. 3n), as they were present in both samples and did not match sizes for alternatively translated peptides.

**Vase tunicate experimental procedures**. Adult *Ciona robusta* animals were collected from both the Gulf of Naples and M-REP (San Diego CA, USA). Ripe oocytes and sperm were collected surgically and kept separately until in vitro fertilization. We used chemical dechorionation to eliminate the chorion and follicular cells surrounding the eggs. Dechorionated eggs were then in vitro fertilized using sperm from various individuals. Fertilized eggs were washed in Millipore-filtered sea water, and transferred to a solution containing 0.77 M Mannitol and 50–80 mg of the plasmid DNA used for electroporation. Electroporation was carried out in Bio-Rad Gene Pulser 0.4 cm cuvettes, using Gene Pulser II (Bio-Rad). Each experiment was performed at least three times. Embryos were staged according to an standard developmental timeline[62]. The *Esrp* and *Twist-like2* probes were obtained from plasmids contained in the *C. robusta* gene collection release I: *Esrp* (VES102_K09) and *Twist-like2* (R1CiGC11k01). A DIG-labeled probe for *Esrp* and a fluorescein-labeled probe for *Twist-like2* were transcribed with Sp6 RNA polymerase (Roche) and purified with the RNeasy Mini Kit (Qiagen). Single and double fluorescent ISH was performed as follows[63]: embryos were fixed at the different developmental stages for 2 h in 4% MEM-PFA (4% paraformaldehyde, 0.1 M MOPS pH 7.4, 0.5 M NaCl, 1 mM EGTA, 2 mM MgSO$_4$, 0.05% Tween 20), then rinsed in cold phosphate-buffered saline (PBS) and stored in 75% ethanol at −20 °C. Embryos were rehydrated, equilibrated in TNT (100 mM Tris pH 7.5, 150 mM NaCl, 0.1% Tween 20) and blocked in TNB (0.5% Roche blocking reagent in 100 mM Tris pH 7.5, 150 mM NaCl) for 1–2 h. Anti-FLUO-POD and anti-DIG-POD (Roche) antibodies were diluted 1:1000 in TNB and incubated with embryos overnight at 4 °C. Embryos were washed extensively in TNT. For tyramide signal amplification, Cy3- and Cy5-coupled TSA Plus reagent (Perkin Elmer) was diluted 1:100 in amplification diluent and added to embryos for 20–40 min at room temperature. For antibody staining, embryos were fixed in 4% MEM-PFA for 30 min, rinsed several times in PBT (PBS with 0.1% Tween 20) and incubated with anti-Cherry (Mouse mAb, Roche, 1:500) with 2% normal goat serum in PBT at 4 °C overnight. Embryos were washed in PBT and then incubated with donkey anti-mouse secondary antibody (1:1000) coupled to Alexa Fluor 488 (Life Technologies) in PBT with 2% normal goat serum for 2 h at room temperature, then washed in PBT. A minimum of 15 embryos of the same stage was used to evaluate expression patterns.

To generate the *Twist > Esrp* construct, we cloned a genomic region upstream *Twist-like1* reported to be active in the mesenchymal lineage[23]. As transcripts for the predicted full-length *Esrp* locus were not detected using a mix of embryonic *Ciona* cDNAs, we cloned the most highly expressed isoform during development for overexpression experiments. This isoform lacks the exonuclease domain at the N-terminus as a result of splice-leader trans-splicing, but contains all RNA-binding domains. Sequences of the primers used in this section are available in Supplementary Table 2.

**Amphioxus experimental procedures**. Adult *Branchiostoma lanceolatum* animals were collected in Banyuls (France) and Mataró (Catalonia). Spawning of reproductive individuals was done as previously reported[64]. Embryos were cultured in filtered sea water at 17 °C and fixed in 4% PFA in MOPS buffer overnight at 4 °C. WMISH was performed with digoxigenin labeled RNA probes for *Esrp*[65].

Hybridization temperature was 65 °C, and antibodies were incubated for 3–4 h, followed by overnight washes in MABT buffer (100 mM maleic acid, 150 mM NaCl, 0.1% Tween-20, pH 8) to reduce background. Detection was done with alkaline phosphatase-conjugated anti-digoxigenin (DIG) antibody. BM Purple (Sigma) was used as chromogenic substrate. A minimum of 15 embryos of the same stage was used to evaluate expression patterns. Sequences of oligonucleotides used in this section are found in Supplementary Table 2.

**Sea urchin experimental procedures**. Adult *Strongylocentrotus purpuratus* specimens were obtained from the Kerckhoff Marine Laboratory, California Institute of Technology, Pasadena CA, USA. Spawning was induced by vigorous shaking of the animals, and embryos were cultured at 15 °C in diluted (9:10) stilled water with 0.22 μm filtered Mediterranean sea water. Embryos were fixed in 4% PFA in MOPS buffer. RNA probes were differentially labeled with digoxigenin (*Esrp*) and fluorescein (*Hnf6*, *Gcm*) in double assays. Colorimetric and fluorescent WMISH and immunohistochemistry with anti-acetylated tubulin antibody were performed according to previous reports[30]. For *Hnf6* and *Gcm* WMISH assays, we used previously published probes[29,66]. A minimum of 15 embryos of the same stage was used to evaluate expression patterns. To knockdown *Esrp*, two different morpholinos were designed, one targeting the single annotated translation start site (5′-CCAGATAGTTAAACGCCATTTTTCC-3′), and the other the 3′ splice site of the third exon in order to induce intron retention (5′-AGTTGTCAAGCTGC-GAAAATGATGA-3′). Morpholinos were obtained from GeneTools and injected (about 6–8 pl per injection) in zygotes at a 300 μM concentration in the presence of 0,12 M KCl. Sequences of primers used to clone the *Esrp* probe and test intron 2 retention are provided in Supplementary Table 2.

**RNA extraction library preparation and sequencing**. RNA extractions were performed using the RNeasy Qiagen Mini Kit (for zebrafish embryos) or the RNaquous (Ambion) (for sea urchin embryos). Trizol extraction (Invitrogen) was employed to obtain total RNA from adult amphioxus tissues according to manufacturer's instructions. All RNA samples were subjected to DNAse treatment. First-strand cDNAs for RT-PCR assays were generated using SuperScript III Reverse Transcriptase (Invitrogen). Libraries for Illumina high-throughput RNA sequencing were produced from poly-A selected RNA as described by the manufacturer, and Illumina HiSeq2500 machines in high yield mode were used for sequencing. Two replicates for DMUT fish embryos and age-matched controls at 5 d.p.f. (the earliest the genotype could be unambiguously identified by eye), and two replicates of splicing morpholino-treated sea urchin embryos at 24 h.p.f. and matched controls were sequenced at high depth to ensure sufficient read coverage over the exon–exon junctions, producing an average of ~145 and ~80 million 125-nucleotide paired-end reads per zebrafish and sea urchin sample, respectively. For human, we used RNA-seq from previous studies[33,34], including one replicate of a *ESRP1* mRNA knockdown (KD) in PNT2 cell line, three replicates of *ESRP1* and *ESRP2* knockdowns in H358 cells, and a replicate of overexpression (OE) of *ESRP1* in MB231 cell line. For mouse, we used previously published RNA-seq data for two replicates of embryonic epidermis from *Esrp1* and *Esrp2* DKO and control mice at the E18.5 stage[15]. Mapping statistics and other features for all RNA-seq samples are provided in Supplementary Table 3.

**Differential gene expression analysis**. Gene expression levels for zebrafish and sea urchin were quantified from RNA-seq data using the cRPKM metrics[67], which employs a single transcript per gene as reference and performs a length correction to account for non-uniquely mapped positions. We used Ensembl version 80 as zebrafish gene annotation (Zv10 assembly) and Spur-v3.1 for sea urchin, retrieved from EchinoBase (previously SpBase; http://www.echinobase.org/). To identify differentially expressed genes in *Esrp* loss-of-function embryos in each species, we first filtered out genes that did not have: (i) a minimal expression of cRPKM > 2 in both replicates in at least one condition (control or loss-of-function); and (ii) at least 50 raw reads supporting expression in at least one sample. For the remaining genes, we required a minimum fold difference of 2 (for zebrafish) or 1.5 (for sea urchin) between the average expression levels in the two conditions, and a fold difference of at least 1.5 (for zebrafish) or 1.2 (for sea urchin) between all pairwise comparisons between each group's replicates.

**Identification of *Esrp*-dependent exons from RNA-seq data**. To quantify all major types of AS from RNA-seq data, we implemented vast-tools[68,69] (https://github.com/vastgroup/vast-tools) for zebrafish and sea urchin. This software uses different modules to identify and quantify simple and complex exon skipping events, intron retention and alternative 5′ and 3′ splice site choices. It has been extensively used to identify differentially spliced AS events in human, mouse, chicken, and planarians, providing high validation rates in RT-PCR assays[47,69–71]. Associated VASTDB files to run vast-tools gene expression and AS analyzes on zebrafish (species key "Dre") and sea urchin (species key "Spu") can be downloaded at http://vastdb.crg.eu/libs/vastdb.dre.10.03.17.tar.gz (Dre) and http://vastdb.crg.eu/libs/vastdb.spu.10.03.17.tar.gz (Spu). vast-tools was also used to quantify AS from human and mouse RNA-seq samples. For each AS event and processed sample in each species, vast-tools provided a table with the percent of alternative sequence inclusion (using the metric 'Percent Splice In', PSI) and a score related to

the number of reads supporting this PSI[69] (N < VLOW < LOW < OK < SOK; see https://github.com/vastgroup/vast-tools for further details).

Next, to identify differentially spliced exons in each organism, we performed the following steps. For zebrafish, sea urchin and mouse we required that a given AS event had sufficient read coverage (score VLOW or higher in the *vast-tools* output) in all compared samples (two control and two loss-of-function replicates). Then, we required that: (i) a minimum absolute PSI change (ΔPSI) between the averages of 15 (for zebrafish and mouse) or 10 (for sea urchin); and (ii) a minimum ΔPSI of 5 between all pairwise comparisons of control and *Esrp* loss-of-function samples. For IR, we further required that the binomial test for read imbalance[70] was not significant ($p < 0.05$) in any of the samples. For human, since we used three different sources, we produced a combined data set consisting of AS events that were differentially spliced (|ΔPSI| > 15 between the control and KD averages, and | ΔPSI| > 5 for all pairwise replicate comparisons in the same direction) in either of the following groupings: (i) the three replicate pairs of the H358 KD experiment; (ii) the PNT2 KD single-replicate experiment and a merged sample of the H358 KD replicates (i.e. pulling all the reads from the three replicates together in each condition to increase read coverage); (iii) the PNT2 KD and MB231 *ESRP1* overexpression single-replicate data (the ΔPSI values should be in the opposite direction); and (iv) the merged H358 KD data set and the MB231 *ESRP1* overexpression data set (again, with ΔPSI values in the opposite direction). All these comparisons were performed by the *vast-tools* module "compare", using the following parameters: --min_dPSI 15 --min_range 5 --p_IR. Finally, cassette exons in a given species that had a homologous counterpart detected as *Esrp*-dependent in another species but that did not pass the initial read coverage filters were re-evaluated and considered *Esrp*-dependent if: (i) only one of the replicates had low coverage values and the total number of reads supporting the PSI was at least two in that sample, and (ii) their PSI values fulfilled the cutoffs described above for the corresponding species.

**Gene ontology analysis**. We used PANTHER statistical overrepresentation test (version 11.1 released 24-10-2016; http://www.pantherdb.org/) with default settings to investigate functional gene enrichment for all species. Sea urchin gene identifiers from v3.1 (starting with WHL22) were converted to SPU identifiers using the conversion table provided by EchinoBase. For differentially expressed genes, we used as background all genes that passed the same initial quality filters (minimal expression of cRPKM > 2 in both replicates of at least one condition and a minimum of 50 raw reads in at least one sample), corresponding to 10,666 genes for zebrafish and 13,574 for sea urchin. For differentially spliced exons, we used raw *p*-values and employed as background all multiexonic genes that fulfilled the same read coverage criteria described above. In total, this corresponded to 11,140 genes in human, 8,501 in mouse, 14,939 in zebrafish and 10,370 in sea urchin. Next, to compare enrichment of specific GO categories among all four species, we selected those categories that were significantly over-represented in at least one species (*p*-value <0.05), and tested enrichment (ratio observed vs. expected) in the raw PANTHER outputs. GO categories that had a ratio >1.3 in all four species were selected and plotted in Supplementary Fig. 6.

**RT-PCR validations and quantitative fluorescent PCRs**. Fluorescent PCRs were performed using forward oligonucleotides marked with a HEX fluorophore reporter (Sigma), with emission at 556 nm. Capillary electrophoresis was performed and analysis and quantification of the amplicons was made with GeneScan software. 500Rox from AppliedBiosystems (Life Technologies) was used as a standard size marker. Sequences of oligonucleotides used in RT-PCRs are provided in Supplementary Table 2.

**Homologous exon clusters**. The clusters of homologous exons in coding regions were assembled according to the following steps. First, clusters of homologous human, mouse, zebrafish, and sea urchin coding genes were built with information from OMA[72], Multiparanoid[73], and BlastP, using the longest protein isoform for each protein-coding gene. For OMA and Multiparanoid, a pair of genes was considered to be homologous if they belonged to the same gene cluster. For BlastP, genes were related if they were in the first three hits in a reciprocal manner between pair of species. Gene clusters were built from orthologous gene pairs that were supported at least by 2 out of the 3 aforementioned approaches, and employing a subsequent guilt-by-association approach. In parallel, aminoacid sequences of all *Esrp*-dependent exons were mapped to all protein isoforms of the harboring genes to obtain a non-redundant set of proteins for each species that contain all *Esrp*-dependent exons. Then, those protein sequences were aligned in a pairwise manner to all isoforms of their homologous gene counterparts using MAFFT. Exon-intron structure information was introduced to the resulting alignment, by intercalating intron positions and their corresponding phase[74].

For each *Esrp*-dependent exon in species A, conservation in species B was assessed based on (i) local exon-intron structure, and (ii) exon sequence similarly. For (i), conservation of upstream and/or downstream flanking intron positions in the aligned region was considered positive if introns with identical phase existed in each species with a maximum deviation of 3 residues in the alignment. For (ii), exon sequence conservation was considered significant if the pairwise similarity between the two species was higher than 20%. However, if an exon significantly

aligned with more than one exon in the other species, local realignment with those exon sequences was performed, and the exon with the highest similarity was kept. With this information, a pair of homologous exons was automatically assigned if the pairwise sequence alignment of the alternative exon and at least one flanking constitutive exon was significant, and both flanking intron positions were conserved. Cases were manually evaluated if: (i) the sequence alignment of the alternative exon and at least one constitutive exon was significant, but at least one of the flanking intron positions was not detected automatically as conserved; or (ii) the upstream and downstream intron positions were conserved, but a low similarity score was obtained for the alternative exon. After this classification, for those exons in human, mouse and zebrafish with no detected homologous counterparts, an additional homology search was performed using liftOver software (http://genome.ucsc.edu/cgi-bin/hgLiftOver) followed by manual curation. Homologous exon clusters of *Esrp*-dependent exons were assembled based on these individually identified exon pairs using a guilt-by-association approach (Supplementary Table 1).

**Esrp RNA regulatory maps**. We used the approach described by[15], with slight modifications. In brief, we first extracted the following sequences for each set of exons (*Esrp*-silenced, *Esrp*-enhanced and non-*Esrp*-dependent): 75 nt from the 3′ and 5′ alternative exon ends, 250 nt from the 5′ and 3′ end of both flanking introns, 75 nt from the 5′ end of the upstream exon, 75 nt from the 3′ end of the downstream exon. If exons or introns were shorter, the corresponding sequences were neglected. Then, a 51-nt window was slid along these sequence sets and the positions covered by any of the 12 top GU-rich SELEX-seq motifs for *Esrp*[33] as a fraction of all positions covered by the window was determined and plotted.

**Conservation of sequences associated with Esrp-dependent AS**. To investigate the conservation of exonic sequences and their flanking intronic regions (150 nt upstream and downstream of the exon), we used phastCons[75] data for human from alignments of 46 vertebrate species, focusing on the subset of placental mammals ('46way.placental' files), downloaded from UCSC (http://genome.ucsc.edu/). To smoothen the signal, for each position we plotted the average across the ten consecutive positions. For *Esrp*-dependent exons, only exons with orthologs in mouse and zebrafish were used. As a reference, we used non-*Esrp*-dependent exons with orthologs in mouse.

**Esrp-like motif analysis of Esrp-dependent exons**. To assess whether the presence of *Esrp*-like motifs at expected positions in *Esrp*-dependent exons was associated with an increased regulatory conservation in other species, we first mapped the 12 top SELEX-seq motifs for *Esrp*[33] to the alternative exon, the first and last 250 nt of the two neighboring introns, and the last and first 25 nt of the upstream and downstream exons, respectively. Then, for each *Esrp*-dependent exon in the studied vertebrate species, we defined putative "direct" targets based on the presence of a minimum number of *Esrp*-like motifs in positions expected from a generalized RNA binding protein regulatory map. In particular, we considered two scenarios. In the "Restricted region" scenario, *Esrp*-silenced exons may have motifs in the alternative exon, in the first 6 nt of the downstream intron (overlapping the 5′ splice site) and/or in the last 25 nt of the upstream intron (see scheme in Supplementary Fig. 9). In the case of *Esrp*-enhanced exons, motifs may be in the first 250 nt of the downstream intron (excluding the 5′ splice site). In the "Expanded region" scenario, in addition to those regions, *Esrp*-silenced exons may have motifs in the first 250 nt of the upstream intron, and *Esrp*-enhanced exons in (i) the last 25 nt of the upstream exon together with the upstream intron (excluding the very last 50 nt next to the 3′ splice site), or (ii) the last 250 nt of the downstream intron. In both scenarios, *Esrp*-silenced exons were considered "indirect" when they had no motifs in the upstream intron and alternative exon and were not considered "direct", and *Esrp*-enhanced exons in the downstream intron and in the last 250 nt of the upstream intron and were not considered "direct". *Esrp*-dependent exons not belonging to any category were discarded from the analyzes.

For Supplementary Fig. 9a, for each species, we used different thresholds of number of motifs for an *Esrp*-dependent exon to be considered "direct" (from ≥1 to ≥4 *Esrp*-like motifs, including overlapping motifs), and calculated the percentage of shared regulation in those homologous exons with sufficient read coverage in the other species. For Supplementary Fig. 9b, for each species, we defined "direct" *Esrp*-dependent exons based on the presence of ≥1 or ≥2 *Esrp*-like motifs under the "Extended region" scenario, divided the homologous exons with sufficient read coverage in the other species into "direct" or "indirect" (or discarded) using the same definitions, and calculated the percentage of shared regulation in each category. One-sided Fisher Exact tests comparing "direct" versus "indirect" proportions were used to assess statistical significance in both analyzes.

**Amphioxus Fgfr AS minigenes and cell cultures**. The full-length *Esrp* open reading frame (ORF) from *B. lanceolatum* and *esrp1* ORF from *D. rerio* were amplified from cDNA using iProof High Fidelity Polymerase (Bio-Rad) and cloned into the pcDNA3.1 vector (Thermo Fisher Scientific). To generate the minigenes for the amphioxus *Fgfr* AS event, the *B. lanceolatum* genomic region spanning the AS event was amplified and cloned into pcDNA3.1; the deletion to generate BlaFgfr-ΔIIIx was generated by PCR using iProof polymerase with reversed oligos

in the flanking intronic regions and the WT minigene as template. Human 293T cells were co-transfected with the minigenes plus a vector encoding amphioxus *Esrp*, zebrafish *esrp1* or an empty vector, using Lipofectamine 2000 (Invitrogen) according to manufacturer's instructions. Total RNA was extracted 24 h after transfection using RNeasy Mini Kit (QIAGEN), cDNA was prepared using SuperScript III (Invitrogen) according to manufacturer's instructions. RT-PCRs for the endogenous AS events and the minigenes were performed using oligos annealing to the adjacent constitutive exons. Amphioxus *Fgfr* amplicons obtained from RT-PCRs were sequenced for isoform identification. Oligonucleotide sequences used in RT-PCRs are found in Supplementary Table 2. 293T cells were checked regularly for mycoplasma contamination, with negative results.

**Genome sources and gene annotation**. To understand the origins and evolution of the mutually exclusive exons IIIb and IIIc in the *Fgfr* paralogs of vertebrates, we performed a comparative analysis in non-vertebrate species. First, we searched for *Fgfr* loci and transcripts in phylogenetically key species that have available genomic and transcriptomic information. Transcript isoforms were obtained by tblastn searches using the human *FGFR2* transcripts as query against public transcriptomes or ESTs databases. Genomic sequences were obtained from public genome databases for *Nematostella vectensis* (nemVec1, JGI), *Crassostrea gigas*, (oyster_v9, JGI), *Drosophila melanogaster* (dm6, FlyBase), *Tribolium castaneum* (triCas2, BeetleBase), *Apis mellifera* (apiMel2, Baylor College of Medicine), *Saccoglossus kowalevskii* (Skow_1.1, Baylor College of Medicine), *Strongylocentrotus purpuratus* (Spur_v3.1, EchinoBase), *Ciona intestinalis* (ci2, Aniseed) and *Danio rerio* (danRer10, Ensembl). Sequencing of amplicons from genomic DNA and tissue-specific cDNA was also carried out for *Branchiostoma lanceolatum*. With this information, GeneWise software (http://www.ebi.ac.uk/Tools/psa/genewise/) was employed to annotate gene structures for each species. Finally, to compare intron/exon structures of the homologous regions to vertebrate's *Fgfr* IgIII domain, we performed interspecific multiple protein alignments with MAFFT, which were manually curated using intron position information and as summarized in Fig. 7.

**Data availability**. Raw RNA-seq data were submitted to Gene Expression Omnibus (GSE97267). Human and mouse RNA-seq data were obtained from SRP066789 and SRP011008 (human), and SRP051370 (mouse). *Esrp* translated transcript sequences from *Salpingoeca urceolata, Salpingoeca kvevrii*, and *Mylnosiga Fluctuans* are provided in Supplementary Table 4. All original gels and blots for PCRs and western blots are provided in Supplementary Fig. 11.

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

## Acknowledgments

We thank Eduard Llorente for drawing the metazoan illustrations, Carlos Herrera and Jacobo Cela for helping with the experiments, Daniel Richter for providing access to unpublished choanoflagellate resources, and Hector Escrivà and Stephanie Bertrand for providing access to amphioxus animals. We also thank Jose Luis Gomez-Skarmeta, Ignacio Maeso and James Sharpe for critical reading of the manuscript and helpful suggestions. Animal silhouettes from Fig. 5, Supplementary Figs. 7–9 were obtained from PhyloPic. RNA sequencing was performed at the CRG Genomics facility, and histological sections were done with the help of the Histology Unit. This work has received funding from the European Research Council (ERC) under the European Union's Horizon 2020 research and innovation program (grant agreement No ERC-StG-LS2-637591 to M.I.), the Spanish Ministry of Economy and Competitiveness (grant BFU2014-58908P to J.G.-F, BFU2014-55076-P to M.I., and the 'Centro de Excelencia Severo Ochoa 2013–2017', SEV-2012-0208), and ICREA - Generalitat de Catalunya (Academia Prize to J.G.-F). We acknowledge the support of the CERCA Programme/Generalitat de Catalunya. D.B. held an APIF fellowship from University of Barcelona, Y.M. an EMBO Long Term postdoctoral fellowship (ALTF 1505-2015), C.R. an EMBO long-term fellowship (ALTF 1608-2014), ATM an FPI-SO fellowship.

## Author contributions

D.B. designed the study, performed in vivo and molecular biology experiments, bioinformatic analyzes, and wrote the manuscript; Y.M. performed bioinformatic analyzes; C.R. did tunicate experiments; J.P. did molecular biology experiments; A.T. performed cell culture and molecular biology experiments; R.E. did tunicate experiments; B.A.-C. did amphioxus experiments; L.F. did molecular biology and in vivo experiments; Y.D'A. did molecular biology experiments; A.G. performed bioinformatic analyzes; E.N.-P. performed molecular biology experiments and bioinformatic analyzes; A.R. performed bioinformatic analyzes; C.C. did sea urchin experiments; G.B. did sea urchin experiments; L.A.C. provided resources; E.M. supervised molecular biology experiments; S.D'A. supervised molecular biology and in vivo experiments; A.S. supervised tunicate experiments; F.R. supervised tunicate experiments; M.I.A. performed and supervised sea urchin experiments and wrote the manuscript; J.G.-F. provided resources and supervision; M.I. performed bioinformatic analyzes, provided resources, wrote the manuscript, and designed and supervised the study.

## Additional information

**Competing interests:** The authors declare no competing financial interests.

