## [Peer Review File · Nature Communications]

Reviewers' comments:

Reviewer #1 (Remarks to the Author):

The manuscript by Burguera et al is a multidisciplinary, interphyletic investigation of Esrp-related splicing programs during the development of some deuterostome species. The work is conducted with care and the amount of data presented is generous. It presents a work with an insightful interplay of bioinformatic, organismal/developmental analyses in which some of the observations made on the genomic level is further investigated on one of the species. The data is well placed in the context of available data from mouse.

Overall the manuscript presents a nice set of investigations that are interesting to the broader readership of Nature Communications. I would have tended to a minor revision but some felt over-interpretations regarding the assumed shared cellular roles of Esrp need to be toned down and in a revised version of the manuscript. The authors search for commonalities in the cellular role and function during embryogenesis where there might not be much - and when these are viewed from an evolutionary perspective even any.

See my following points with the topics regarding the discussion at the end.

Branchiostoma:

"in the amphioxus *Branchiostoma lanceolatum*, a cephalochordate species that shares many developmental processes and a general bodyplan with vertebrates[25]."

I suggest to tone down the suggested similarities - many (most) developmental processes are different and create basically a different animal, namely a vertebrate with its unique and numerous novelties. The presented data also shows a far more restricted activity of Esrp than in zebrafish - contrary to this similarity statement. In principle this could be also highlighted in the text.

Echinoderm:

"Ambulacraria, a lineage closely related to chordates,"

This is of course a relative statement and could be better specified, e.g. as "sister group to Chordata." This is not relative and far more informative.

Also, the in-situ hybridization in Figure 4a-c is of poor quality. It would be great if they could be improved so that the reader can orientate herself in the embryo. It is impossible now.

Also the double fluorescent in situ hybridizations are not so insightful when presented as they are. 4d, e are OK, but in 4f the orientation of the embryo is not clear. Is it turned 90 degrees?

Please explain 'chickenpox' phenotype. Its urchin slang.

From "This failure in the complete integration of pigment cells into the ectoderm likely constitutes an impaired mesenchymal-to-epithelial transition." to "In summary, these results demonstrate that Esrp in sea urchin is necessary for a complete integration of pigment cells into its destination epithelium." What is with the other cells that undergo a epithelial-to-mesenchymal transition such as the primary and secondary mesenchyme? It looks like Esrp is expressed asymmetrically - any explanation for this?

Other changes:

"that occurred at the base of the chordate phylum, before the split of its three main subphyla (Fig. 7)." change to "that occurred in the lineage to the Chordata (Fig. 7)"

"which comprises echinoderms and hemichordates phyla" change to "which comprises echinoderms and hemichordates."

Discussion:

For my taste the authors stretch too much the role of *Esrp* in the mesenchymal to epithelial transition across the investigated species. It is clear from the data that *Esrp* is widely expressed, that not all EMT or MET processes involve *Esrp* and that *Esrp* is also active in processes that have nothing to do with cell motility or epithelial integration.

However, overall there is a confirmatory bias of the authors to equalize these and similar cellular processes in their animals. Sentences such as "These data point to similar roles for *Esrp* modulating epithelial integration and/or cell motility properties in vertebrates, although whether any of the observed embryonic phenotypes are due to defects on this type of processes need to be further investigated." are evident for this and are also stay extremely vague and not very informative. Similar case is the following sentence: "Altogether, our results thus suggest that *Esrp* genes provide epithelial-associated features to certain cells involved in diverse developmental processes across species." Such statement could be made about many other genes and if this 'shared' employment is significant for an evolutionary conservation remains very unclear from the presented data in the organisms. I think the contrary is true and the differences observed between sea urchin and chordates are support for this statement.

I suggest to exclude the speculation from the discussion and keep it in some minor statements in the results section and restrict the evolution of *Esrp* function to only two levels, namely the organismic and molecular level.

Reviewer #2 (Remarks to the Author):

This study explores the role of Epithelial Splicing Regulatory Protein (*Esrp*) and its impact on alternative splicing, including master regulators such as the FGFR gene family in cellular differentiation. It provides important insights into the evolution of ESRP as a transcript regulator within vertebrates and highlights its capacity to increase transcript diversity in a species-specific manner. The study is well designed and thorough, and the manuscript is well written. My main advice to the authors would be to go a bit further beyond a descriptive level, and exploit the opportunity to further understand the regulatory evolution using the available data. The authors could improve their definition of an 'unchanged' or 'non-regulated' exon. Moreover, a more thorough analysis of the regulatory motifs would be required to properly define the categories of exons with conserved and non-conserved regulation, as described below.

1. This study identifies 365 *Esrp*-regulated exons in sea urchin, 494 in zebrafish, 254 in mouse and 336 in human. Then it performs overlaps (shown with Venn diagrams) between these groups to distinguish exons into overlapping or non-overlapping between species. However, there is a problem with this analysis, since the lack of overlap between these lists doesn't necessarily mean that the exon is 'unchanged' or 'non-regulated' in one of the species. Just because the exon is not detected by a specific software in a specific RNA-seq performed at a specific time point, or a specific tissue, it doesn't mean that this exon is not regulated in a certain species. The non-overlap could result from differences in variance or quality of data between species – I assume mouse has deeper coverage and more higher quality data available? As a result, the proportion of false positive or false negative detection could vary between data in each species. Moreover, the developmental stages and tissue sections are certainly not perfectly analogous, and thus a non-overlapping exon could be changed at a different time point in development. To overcome these issues, the authors could consider the following analyses:

- To avoid the effects of data variance when performing comparisons across species, the overlap

between species would not be done only by looking at exons that passed a significance threshold in each species, but rather the PSI values as defined by RNAseq could be directly compared. For example, orthologues are defined for all exons identified in any of the species. If an exon is found in only a single species (i.e., no orthologues), then it is removed from the analysis (since no meaningful comparisons can be made). For the remaining exons, PSI value is calculated in each species, and then PSI values are compared across tissues. A threshold is defined for the PSI value that is considered to represent a meaningful change, and exons crossing this value are considered changed, while remaining ones are not. Then the comparison is made between species to see which exons pass this threshold (and change in the same direction) in multiple species, and which ones don't. This is just an example, any variant of this approach would be fine.

- When building of homologous exon clusters, did the algorithm check that splice sites exist in all of the homologous exons? I couldn't completely understand this from the methods. What is considered under the 'local exon-intron structure'? This is a bit unclear.

2. A previous study showed that ESRP binding motifs (especially UGGUGG) are enriched flanking ESRP regulated exons in mouse epidermis (<https://lens.elifesciences.org/08954>). This was done by analyzing the top 12 GU-rich ESRP motifs that were previously identified by the SELEX-Seq (<http://mcb.asm.org/content/32/8/1468>) to identify the RNA binding map of the ESRPs around the differentially spliced exons. Recently, the RNA Bind-n-Seq for ESRP1 has also become available from ENCODE, which can complement the motif analysis. The maps of ESRP motifs agreed with the pattern seen for Nova and most other splicing regulators, with enriched motifs upstream or within the silenced exons, and downstream of enhanced exons. Thus, it is clearly possible to identify regulatory motifs at relevant positions around the affected exons to define if an exon is a direct and indirect ESRP-regulated event. The RNA binding maps and the regulatory maps are usually highly conserved, so it should not be a problem to use same motifs for analysis across species. This has been shown for Nova protein by the authors, and also by other studies (<https://www.ncbi.nlm.nih.gov/pubmed/20921232>). Thus, I would propose that the authors attempt some or all of the following analyses:

- Analyse the RNA binding map of ESRP binding motifs around the 365 exons identified in sea urchin, 494 in zebrafish, 254 in mouse and 336 in human using the same or similar approach as Fig 5C in <https://lens.elifesciences.org/08954>. This will also assess the quality of data for each species, as the extent of motif enrichment reflects the % of direct ESRP-regulated events that were detected.

- Divide exons in each species into candidate direct and indirect ESRP-regulated events based on the presence of ESRP binding motif at expected positions. Compare how likely each group is to be conserved in another species. Are direct events more likely conserved? One would expect this, since the indirect events are more likely a result of technical noise or species & tissue-specific response to perturbed ESRP function.

- Evaluate the conservation of ESRP binding motifs that are present around the direct ESRP-regulated events – if the motif is conserved in another species, did RNAseq also confirm the conserved regulation?

- It is known that alternative exons generally have more highly conserved flanking intronic sequence. It would be of interest to know if the exons with conserved ESRP regulation have increased conservation of their flanking intronic sequence compared to the remaining non-conserved ESRP-regulated exons

Reviewer #3 (Remarks to the Author):

The manuscript by Burguera et al. attempts a detailed analysis of expression of ESRPs during embryonic morphogenetic processes in several different phylogenetic clades and tries to find a correlation of their roles at cellular and molecular levels during evolution. The study has characterized the expression profile of ESRP during development in multiple organisms; however, similar patterns

have been observed in previously published studies (Warzecha et al. 2009, Mol Cell; Bebee et al. 2015, eLIFE; Bhate et al. 2015, Nat Commun; Bebee et al. 2016 Dev. Dyn). Although the investigators have performed a diverse set of experiments, the study is highly descriptive with little mechanistic understanding. Most conclusions are not novel and do not provide significant new insights either into ESRPs function or evolution. Several of the findings are expected and have been reported in previous studies.

Major Comments:

1. The characterization of the mutant alleles of ESRP1 and ESRP2 is unsatisfactory. It is not clear, whether the ESRP1 mutant results in a complete loss of protein or makes a truncated form. If indeed, the ESRP1 mutant results in a truncated protein, it might still have activities independent of splicing. A western blot is needed to confirm this. It would also be nice if the authors stuck to one manipulation of ESRP expression with all of the organisms tested (i.e. removing expression by knockout or knockdown) so the results would be directly comparable. For example in Figure 2, ectopically expressing ESRP in all mesenchymal cells seems a bit out of place. If the goal were to show ESRP's conserved roles in morphogenesis, it would be better to remove ESRP rather than expressing it in a subset of cells where it is normally absent. This would allow direct comparison between the zebrafish knockout and the sea urchin knockdown via morpholinos.
2. The authors conclude that the ESRP dependent splicing program is not conserved across widely separated phyla. This finding is obvious- as the evolutionary distance increases; conservation of gene expression networks tends to decrease. Nonetheless, the finding that ESRP exerts nearly identical functions with almost no conservation in target genes is intriguing and should be explored further. The results as shown only skim the surface and do not provide any biological insights into the molecular functions of ESRPs.
3. The authors describe a phylum-specific AS event in the FGFR family required for normal developmental programs whose gene structure has evolved across metazoans. It is not really unexpected that ESRPs regulate this exon in other vertebrate species as well. The authors provide evidence for the evolution of flexible splicing programs in ESRP-dependent processes; however, it is at times unclear what the goals of many of the experiments are. Early on in the manuscript, it appears that the authors want to investigate the phenotype of ESRP DKO in Zebrafish. It was not until the end of the paper that the authors presented their work as a case for the evolution of flexible splicing programs. The study may be better organized under a single theme.

Minor Comments:

1. The single mutant of *Esrp1* and *2* did not show developmental defects in comparison to DKO in Figure 1. Does the expression of the other ESRP paralogue increase as a compensating mechanism?
2. For Figure 3, some manipulation of ESRP expression in *Branchiostoma lanceolatum* should have been carried out to support the claim of conserved function. As it currently sits, it does not belong as main figure.
3. In figure 4, it is hard to see much of a difference between the control and morpholino-treated embryos between panels J and K. The legend for these panels and their reference in the text states that abnormally long cilia ("hairy" phenotype) can be seen in panel K, but it does not look like it. This is shown in panels N and O. For panel R, the authors do not offer an explanation of what differentiates between a strong or mild hairy phenotype.
4. Also, in Figure 4, the authors use morpholinos to knockdown ESRP in sea urchin zygotes. However, it is unclear how intron 2 retention fails to produce functional ESRP protein. Also, why was translation inhibition used as a control for this knockdown?
5. The authors show in Figure 6c that zebrafish ESRP1 only produced mild changes in exon IIIb inclusion. Why is this the case? One would expect equivalent changes unless the presence of the IIIx exon is altering its splicing efficiency in some way. Please elaborate.
6. Figure 7 shows the variability of the IgIII domain structure of FGFR, however, it does not show the

evolutionary change of that structure through time as the authors suggest. Altering the figure in a way that clearly outlines the evolutionary timeline that the authors have explained in the text would enhance the readability of the manuscript.

7. Please explain why the RNAseq experiment on Zebrafish DKO and WT larvae was performed at 5 dpf, instead of before or later when survival rates go down? Also, was there a reason to choose Sp10-injected embryos, but not TrMo-injected embryos, for RNA-seq?

8. In figure 2m-n, Twist>GFP may be a better control than Tyrp1/2a>2xGFP.

9. The language within the paper is confusing/difficult to follow at many places with a lot of unnecessary use of jargon. It also feels like the description of the results is stretched, making it unnecessarily lengthy. Therefore, the authors may want to modify and present results in a better and concise manner, changing figures to include pertinent information and improving the overall flow of the paper.

Point-by-point response to reviewers

Reviewer #1 (Remarks to the Author):

The manuscript by Burguera et al is a multidisciplinary, interphyletic investigation of *Esrp*-related splicing programs during the development of some deuterostome species. The work is conducted with care and the amount of data presented is generous. It presents a work with an insightful interplay of bioinformatic, organismal/developmental analyses in which some of the observations made on the genomic level is further investigated on one of the species. The data is well placed in the context of available data from mouse.

Overall the manuscript presents a nice set of investigations that are interesting to the broader readership of Nature Communications. I would have tended to a minor revision but some felt over-interpretations regarding the assumed shared cellular roles of *Esrp* need to be toned down and in a revised version of the manuscript. The authors search for commonalities in the cellular role and function during embryogenesis where there might not be much - and when these are viewed from an evolutionary perspective even any.

R: We thank the Reviewer for his/her appreciation of our work. We have now toned down the interpretations regarding the shared cellular roles of *Esrp*, and have been more explicit about plausible evolutionary implications of our findings (see more details below).

See my following points with the topics regarding the discussion at the end.

Branchiostoma:

“in the amphioxus *Branchiostoma lanceolatum*, a cephalochordate species that shares many developmental processes and a general bodyplan with vertebrates[25].” I suggest to tone down the suggested similarities - many (most) developmental processes are different and create basically a different animal, namely a vertebrate with its unique and numerous novelties. The presented data also shows a far more restricted activity of *Esrp* than in zebrafish - contrary to this similarity statement. In principle this could be also highlighted in the text.

R: We have reworded this sentence as suggested by the Reviewer. We have also added a mention to the more restricted expression of *Esrp* in amphioxus.

Echinoderm:

“Ambulacraria, a lineage closely related to chordates,” This is of course a relative statement and could be better specified, e.g. as “sister group to Chordata.” This is not relative and far more informative.

R: The change has been made.

Also, the in-situ hybridization in Figure 4a-c is of poor quality. It would be great if they could be improved so that the reader can orientate herself in the embryo. It is impossible

now.

Also the double fluorescent in situ hybridizations are not so insightful when presented as they are. 4d, e are OK, but in 4f the orientation of the embryo is not clear. Is it turned 90 degrees?

R: We have now improved the clarity of the figure by adding axis for orientation to the complex panels as well as outlining the embryo structures in the fluorescent in situ hybridizations. Moreover, we have added additional information to the figure legend.

Please explain ‘chickenpox’ phenotype. Its urchin slang.

R: In fact, we had described this phenotype here for the first time. Since the naming of the phenotype is certainly not required, we have now simply removed it and described the phenotype in more detail to avoid confusions.

From “This failure in the complete integration of pigment cells into the ectoderm likely constitutes an impaired mesenchymal-to-epithelial transition.” to “In summary, these results demonstrate that *Esrp* in sea urchin is necessary for a complete integration of pigment cells into its destination epithelium.”

What is with the other cells that undergo a epithelial-to-mesenchymal transition such as the primary and secondary mesenchyme? It looks like *Esrp* is expressed asymetrically - any explanation for this?

R: Our results indicate that *Esrp* is involved in the mesenchymal-to-epithelial transition (MET) performed by aboral non-skeletogenic mesodermal cells during gastrulation. Both the primary mesenchyme, which later develops into the larval skeleton, and other populations of the secondary mesenchyme, such as the myogenic lineage, undergo the opposite cellular behaviour (i.e. epithelial-to-mesenchymal transitions, EMT) to detach from the surrounding endomesoderm. *Esrp* is not expressed in these cells when they undergo EMT and, consistently, they do not show any defect upon *Esrp* knockdown.

Regarding the second question, only the aboral secondary mesenchyme cell lineage experiences a MET during gastrulation, providing a coherent explanation for the asymmetrical expression of *Esrp*. We have now made this reasoning more explicit in the text.

Other changes:

“that occurred at the base of the chordate phylum, before the split of its three main subphyla (Fig. 7).” change to “that occurred in the lineage to the Chordata (Fig. 7)”

“which comprises echinoderms and hemichordates phyla” change to “which comprises echinoderms and hemichordates.”

R: Both suggestions have been incorporated into the manuscript. Thank you.

Discussion:

For my taste the authors stretch too much the role of *Esrp* in the mesenchymal to

epithelial transition across the investigated species. It is clear from the data that *Esrp* is widely expressed, that not all EMT or MET processes involve *Esrp* and that *Esrp* is also active in processes that have nothing to do with cell motility or epithelial integration. However, overall there is a confirmatory bias of the authors to equalize these and similar cellular processes in their animals. Sentences such as “These data point to similar roles for *Esrp* modulating epithelial integration and/or cell motility properties in vertebrates, although whether any of the observed embryonic phenotypes are due to defects on this type of processes need to be further investigated.” are evident for this and are also stay extremely vague and not very informative. Similar case is the following sentence: “Altogether, our results thus suggest that *Esrp* genes provide epithelial-associated features to certain cells involved in diverse developmental processes across species.” Such statement could be made about many other genes and if this ‘shared’ employment is significant for an evolutionary conservation remains very unclear from the presented data in the organisms. I think the contrary is true and the differences observed between sea urchin and chordates are support for this statement. I suggest to exclude the speculation from the discussion and keep it in some minor statements in the results section and restrict the evolution of *Esrp* function to only two levels, namely the organismic and molecular level.

R: We agree that some of the interpretations regarding commonalities of cellular behaviours may have been due to confirmatory biases, and we have done our best to remove such biases and other speculations on the cellular level, and discuss only our functional results in a comparative and evolutionary framework.

Just to clarify, however, we did not intend to claim that *Esrp* is mainly associated with EMT-METs or that *Esrp* cellular functions are most often shared, and we apologize if this has caused confusion. Similarly, we certainly agree that *Esrp* genes are playing a wide variety of roles in the different studied organisms (as our Title attests), and we are now more explicit about this. Our aim was to discuss some commonalities we have observed across species, which include a conserved expression in the non-neural ectoderm and a set of phenotypes in some of the studied lineages that seem consistent with defects on specific (not all) METs. We have now explicitly acknowledged that *Esrp* is involved only in certain METs, and we only briefly discuss plausible Evo-Devo inferences regarding the deuterostome ancestors and not shared cellular roles.

Reviewer #2 (Remarks to the Author):

This study explores the role of Epithelial Splicing Regulatory Protein (Esrp) and its impact on alternative splicing, including master regulators such as the FGFR gene family in cellular differentiation. It provides important insights into the evolution of ESRP as a transcript regulator within vertebrates and highlights its capacity to increase transcript diversity in a species-specific manner. The study is well designed and thorough, and the manuscript is well written. My main advice to the authors would be to go a bit further beyond a descriptive level, and exploit the opportunity to further understand the regulatory evolution using the available data. The authors could improve their definition of an ‘unchanged’ or ‘non-regulated’ exon. Moreover, a more thorough analysis of the regulatory motifs would be required to properly define the categories of exons with conserved and non-conserved regulation, as described below.

R: We thank the Reviewer for his/her appreciation of our work, and for the thorough and constructive suggestions on how to improve and extend our analyses.

1. This study identifies 365 *Esrp*-regulated exons in sea urchin, 494 in zebrafish, 254 in mouse and 336 in human. Then it performs overlaps (shown with Venn diagrams) between these groups to distinguish exons into overlapping or non-overlapping between species. However, there is a problem with this analysis, since the lack of overlap between these lists doesn’t necessarily mean that the exon is ‘unchanged’ or ‘non-regulated’ in one of the species. Just because the exon is not detected by a specific software in a specific RNA-seq performed at a specific time point, or a specific tissue, it doesn’t mean that this exon is not regulated in a certain species. The non-overlap could result from differences in variance or quality of data between species – I assume mouse has deeper coverage and more higher quality data available? As a result, the proportion of false positive or false negative detection could vary between data in each species. Moreover, the developmental stages and tissue sections are certainly not perfectly analogous, and thus a non-overlapping exon could be changed at a different time point in development.

R: We thank the Reviewer again for his/her thoughtful insights on the main technical and conceptual challenges associated with evolutionary comparisons of exon targets. Certainly, the fact that the samples for each species are necessarily heterogeneous complicates direct comparisons. We have followed his/her suggestions to improve our comparisons and better present the results (see below). In addition, we have added the following cautionary note: “Finally, it should be noted that additional transcriptomic data from other tissues or developmental stages may increase the fraction of shared *Esrp*-dependent exons detected among species”.

To overcome these issues, the authors could consider the following analyses:

- To avoid the effects of data variance when performing comparisons across species, the overlap between species would not be done only by looking at exons that passed a significance threshold in each species, but rather the PSI values as defined by RNAseq could be directly compared. For example, orthologues are defined for all exons identified in any of the species. If an exon is found in only a single species (i.e., no orthologues), then it is removed from the analysis (since no meaningful comparisons can be made). For the remaining exons, PSI value is calculated in each species, and then

PSI values are compared across tissues. A threshold is defined for the PSI value that is considered to represent a meaningful change, and exons crossing this value are considered changed, while remaining ones are not. Then the comparison is made between species to see which exons pass this threshold (and change in the same direction) in multiple species, and which ones don't. This is just an example, any variant of this approach would be fine.

R: We have indeed performed similar steps in our original analysis to call “shared” *Esrp*-dependent exons, but we agree that extra care should be taken when calling “non-shared” exons. We have now performed the following steps. First, as suggested by the Reviewer, we have excluded the exons with no homologous in any species from the Venn diagram (indicated as “NH” in Fig. 5b), and provide a pairwise comparison of the percent of exons with homologous in each species (Supplementary Fig. 8b). Second, for those *Esrp*-dependent exons that have homologs in at least another species, we removed from the Venn diagram those that do not have enough read coverage in all species in which we can find homologs (“NC” in Fig. 5b). Then, we directly compared the changes in inclusion level after *Esrp* depletion, and required that they were above a given threshold and went in the same direction. Therefore, within the level of confidence allowed by our datasets, exons that are called as “non-shared” in our Venn diagram are either not *Esrp*-dependent or do not have a homolog in the other species. To complement this information, we also provide a pairwise comparison of the percentage of *Esrp*-regulated exons with homologs and sufficient read coverage in other species that have shared *Esrp* regulation in that species (Supplemental Fig. 8c). We believe these analyses offer more detailed information and more faithfully reflect the different levels of evolutionary conservation of *Esrp*-dependent alternative splicing regulation.

- When building of homologous exon clusters, did the algorithm check that splice sites exist in all of the homologous exons? I couldn't completely understand this from the methods. What is considered under the ‘local exon-intron structure’? This is a bit unclear.

R: Yes, presence of splice sites is taken into account. Moreover, only exons with transcriptional evidence are included in the clusters.

With “local exon-intron structure” we refer to the conservation of the intron positions surrounding the alternative exon. That is, we performed protein alignments for pairs of species and introduced the intron positions (and phases) into these alignments (e.g. see Irimia and Roy, NAR 2008). This allowed us to better assess whether the intron positions that define the alternative exons are ancestral to the two species, and thus whether the alternative exons are true orthologs. We have now provided a more detailed explanation in the Methods section.

2. A previous study showed that *Esrp* binding motifs (especially UGGUGG) are enriched flanking *Esrp* regulated exons in mouse epidermis (<https://lens.elifesciences.org/08954>). This was done by analyzing the top 12 GU-rich *Esrp* motifs that were previously identified by the SELEX-Seq (<http://mcb.asm.org/content/32/8/1468>) to identify the RNA binding map of the *Esrps*

around the differentially spliced exons. Recently, the RNA Bind-n-Seq for ESRP1 has also become available from ENCODE, which can complement the motif analysis. The maps of *Esrp* motifs agreed with the pattern seen for Nova and most other splicing regulators, with enriched motifs upstream or within the silenced exons, and downstream of enhanced exons. Thus, it is clearly possible to identify regulatory motifs at relevant positions around the affected exons to define if an exon is a direct and indirect *Esrp*-regulated event. The RNA binding maps and the regulatory maps are usually highly conserved, so it should not be a problem to use same motifs for analysis across species. This has been shown for Nova protein by the authors, and also by other studies (<https://www.ncbi.nlm.nih.gov/pubmed/20921232>). Thus, I would propose that the authors attempt some or all of the following analyses:

- Analyse the RNA binding map of *Esrp* binding motifs around the 365 exons identified in sea urchin, 494 in zebrafish, 254 in mouse and 336 in human using the same or similar approach as Fig 5C in <https://lens.elifesciences.org/08954>. This will also assess the quality of data for each species, as the extent of motif enrichment reflects the % of direct *Esrp*-regulated events that were detected.

R: We used the top 12 GU-rich SELEX motifs for *Esrp* to build RNA binding maps for each species, following the approach described by Bebee et al (Elife 2015). We observed enrichment of *Esrp* motifs above the background for all species in expected locations (Supplementary Fig. 8a), although results for sea urchin were less clear, likely reflecting a lower number of direct targets in this species due to the use of RNA-seq from knockdowns of whole embryos, which have higher tissue heterogeneity and are thus more susceptible to indirect effects upon development perturbations.

- Divide exons in each species into candidate direct and indirect *Esrp*-regulated events based on the presence of *Esrp* binding motif at expected positions. Compare how likely each group is to be conserved in another species. Are direct events more likely conserved? One would expect this, since the indirect events are more likely a result of technical noise or species & tissue-specific response to perturbed *Esrp* function.

R: We thank the reviewer for his/her nice suggestions, which we have developed. However, we would first like to point out that, as the Reviewer is most certainly aware, calling direct/indirect targets based merely on the presence/absence of sequence motifs is a complex matter, especially given that the ESRP binding motif has a relatively weak sequence consensus (compared to other RNA-binding proteins such as RBFOX, for instance), resulting in a larger fraction of false positive calls. Moreover, the positional effect is indeed rather complex when examining exons on a one-to-one basis. For example, similar to silenced exons, enhanced exons may show binding of the regulator in the upstream intron (e.g. Ule et al, Nature 2016; likely when it binds upstream of the branch point), and downregulation may be due to binding in the exon and downstream intron (when close to the 5' splice site).

With these caveats in mind, we have performed the analyses using two scenarios for the position of *Esrp*-like motifs (Supplemental Fig. 9a): a “restricted” one, in which only sequences from the alternative exon and the neighbouring intronic regions were considered, and an “extended” one, in which all enhancer and silencer positions defined in the canonical RNA regulatory map were considered. Furthermore, we have applied

different levels of stringency in the calls of direct exons, using ≥ 1 to ≥ 4 SELEX-derived *Esrp*-like hexamers in the expected positions. As predicted by the reviewer, we observed that exons with *Esrp*-like motifs tend to have higher conservation of their regulation (Supplementary Fig. 9a). However, due to the low sample size many comparisons against the “non-direct” exons were not statistically significantly different.

- Evaluate the conservation of *Esrp* binding motifs that are present around the direct *Esrp*-regulated events – if the motif is conserved in another species, did RNAseq also confirm the conserved regulation?

R: Using a similar rationale as the one described above, we enquired, for each vertebrate species, whether exons with *Esrp*-like motifs in expected positions showed higher levels of regulatory conservation when the orthologs also had *Esrp*-like motifs in equivalent positions. For most (yet not all) cases, we saw indeed higher conservation when motifs were conserved. This analysis has been included as Supplemental Fig. 9b.

- It is known that alternative exons generally have more highly conserved flanking intronic sequence. It would be of interest to know if the exons with conserved *Esrp* regulation have increased conservation of their flanking intronic sequence compared to the remaining non-conserved *Esrp*-regulated exons.

R: We quantified the conservation of flanking intronic sequences of the human *Esrp*-dependent exons with homologous counterparts in mouse and zebrafish species using phastCons scores. We separated those clusters into three sets, depending if they were detected as *Esrp*-dependent in the three organisms, in the two mammals or only in human. Nicely fitting the reviewer's prediction, we found that flanking intronic sequences of exons with shared *Esrp*-dependent regulation among vertebrates or between mammals are indeed more conserved than those human-specific exons as well as the non-regulated background (Supplementary Fig. 8d).

Reviewer #3 (Remarks to the Author):

The manuscript by Burguera et al. attempts a detailed analysis of expression of *Esrps* during embryonic morphogenetic processes in several different phylogenetic clades and tries to find a correlation of their roles at cellular and molecular levels during evolution. The study has characterized the expression profile of ESRP during development in multiple organisms; however, similar patterns have been observed in previously published studies (Warzecha et al. 2009, Mol Cell; Bebee et al. 2015, eLIFE; Bhate et al. 2015, Nat Commun; Bebee et al. 2016 Dev. Dyn). Although the investigators have performed a diverse set of experiments, the study is highly descriptive with little mechanistic understanding. Most conclusions are not novel and do not provide significant new insights either into ESRPs function or evolution. Several of the findings are expected and have been reported in previous studies.

R: We are sorry to read that this Reviewer did not appreciate the interest and novelty of our study. However, we respectfully disagree. First, this is primarily an evolutionary study, and all the papers cited by the Reviewer concern mouse or human data. This underscores the clear existing need to properly investigate *Esrp* beyond mammals. Second, and somewhat surprisingly, the Reviewer says: "similar patterns have been observed in previously published studies". We assume he/she is referring to zebrafish, where we described largely conserved expression and functions for *Esrp* genes compared to mouse. However, this is itself an important result, and it is not necessarily expected that organismal functions of a splicing factor have been conserved for over 450 million years. More importantly, many expression patterns and functional results described in non-vertebrate organisms are simply not conserved with vertebrates, and thus cannot be "not novel" or "expected". Third, we strongly disagree that the study is "highly descriptive"; we provide highly detailed functional, *in vivo*, experiments for three species, something extremely unusual in the literature and that, to our knowledge, has never been done before in the alternative splicing field. Thus, we think that the claim that the conclusions do not provide significant new insights into ESRPs evolution and function is unjustified.

Major Comments:

1. The characterization of the mutant alleles of ESRP1 and ESRP2 is unsatisfactory. It is not clear, whether the ESRP1 mutant results in a complete loss of protein or makes a truncated form. If indeed, the ESRP1 mutant results in a truncated protein, it might still have activities independent of splicing. A western blot is needed to confirm this.

R: Since there were no available antibodies for zebrafish *Esrp1* or *Esrp2*, we ordered custom antibodies to an external company. While the custom anti-*Esrp2* antibody did not work with zebrafish samples despite multiple attempts, the anti-*Esrp1* antibody yielded a strong band of the expected size for WT embryos, in addition to several weaker shorter bands that most likely correspond to unspecific signal (similar bands were not reported for mouse ESRP1). Fortunately, we then tried a commercial antibody against mouse ESRP2 and this showed cross-species recognition, despite the presence of a few unspecific bands (many unspecific bands were observed in mouse with these antibody; see Mizutani et al, Oncogene 2016). In western blots with each antibody, the bands corresponding to endogenous full-length *Esrp* proteins were clearly absent in the

mutant samples (Supplementary Fig. 3n). In the case of *Esrp1*, only faint bands with smaller size were observed that could correspond to a truncated protein in the mutant embryos. Therefore, even if the mutant alleles were translated, they would do so at low levels and as truncated proteins, being at the very least effective loss-of-function knockdowns. Moreover, it is very important to reiterate four other complementary results: (i) the expression of the mutant alleles is highly reduced in the mutant embryos. We had reported this based on RNA-seq in the initial submission, and we have now validated these patterns using qPCRs in single and double mutants (see below); (ii) overexpression of the mutant alleles in human cells showed no effect on alternative splicing; (iii) the developmental defects we describe for zebrafish are highly consistent with those of mouse *Esrp* double KOs, further supporting their specificity; and (iv) relatedly, as in the mouse double KO, we observed full switches in the splicing of all *Fgfr* events, which are known to be strongly dependent on *Esrp* function. Therefore, we are confident that our zebrafish lines are robust loss-of-function models.

However, we fully agree with the Reviewer that it is best to be cautious, particularly given that the exact degree of loss of functionality in our mutants is not relevant for the conclusions of our study. Therefore, we have now referred to our zebrafish lines as mutants, and not KOs, throughout the text, and acknowledge that we refer to a loss-of-function mutant with regards to the splicing regulatory activity and that an effect on other regulatory processes cannot be completely ruled out.

It would also be nice if the authors stuck to one manipulation of ESRP expression with all of the organisms tested (i.e. removing expression by knockout or knockdown) so the results would be directly comparable. For example in Figure 2, ectopically expressing ESRP in all mesenchymal cells seems a bit out of place. If the goal were to show ESRP's conserved roles in morphogenesis, it would be better to remove ESRP rather than expressing it in a subset of cells where it is normally absent. This would allow direct comparison between the zebrafish knockout and the sea urchin knockdown via morpholinos.

R: While we agree that it would be nice to also have loss-of-function data for *Ciona*, in this particular case we were interested in testing a very specific hypothesis about the impact of *Esrp* on a well-characterized process in this organism, given its mutually exclusive expression pattern with *Twist-like2* within the developing mesenchymal lineage. Specifically, we reasoned that lack of silencing of *Esrp* during development in the *Twist-like2*-positive lineage would result in ontogenetic alterations, and, for this purpose, overexpression of *Esrp* under a *Twist* promoter was the appropriate tool. We have now made this reasoning more explicit in the text.

Furthermore, it should be pointed out that performing loss-of-function experiments in *Ciona* is far from trivial. Currently, the best way to achieve this is through overexpression of a dominant negative form of the target gene (not available for *Esrp*), which, in itself, has many caveats. Transgenesis using CRISPR-Cas9 is currently being developed, but it is limited to F0 embryos with low mutation rates and high levels of mosaicism, which makes any robust loss-of-function analysis unfeasible at this point. Therefore, these approaches are not straightforward and would have required a dedicated study on its own, which is beyond the scope of the present MS.

2. The authors conclude that the ESRP dependent splicing program is not conserved across widely separated phyla. This finding is obvious- as the evolutionary distance increases; conservation of gene expression networks tends to decrease. Nonetheless, the finding that ESRP exerts nearly identical functions with almost no conservation in target genes is intriguing and should be explored further. The results as shown only skim the surface and do not provide any biological insights into the molecular functions of ESRPs.

R: There are three related points that we think make these results interesting. First, evolution of splicing regulatory networks has been barely studied, and never with the phylogenetic breadth of this study. Second, although higher divergence is certainly expected between more distant clades, it is important to precisely determine the different degrees of divergence at nested evolutionary distances (i.e. between mammalian, bony vertebrates, chordates and deuterostome clades) to properly understand macroevolutionary patterns. Therefore, while it is obvious that conservation of gene regulatory networks will tend to decrease with evolutionary distance, it is not at all known what the paucity of such decrease is for alternative splicing programs. Third, our results not only revealed a temporal trend, but also how specific exons are recruited into regulatory networks at different phylogenetic distances (i.e. largely by recruiting homologous alternative exons between mammals, but non-homologous, novel exons between phyla).

Regarding the evolution of molecular functions, we now emphasize two main insights as possible explanations regarding the common employment of the *Esrp* gene family in some morphogenetic processes across deuterostomes. First, Gene Ontology analysis detected a significant enrichment of genes related to several shared categories in vertebrates and sea urchin. Second, although we did not identify conserved *Esrp*-dependent exons between sea urchin and vertebrates, we detected homologous targets at the gene level for 21 cases. Thus, a major implication of our results is that, although the specific exons targeted by a splicing factor are flexible, such factors often regulate related cellular pathways and even the same orthologous genes across phyla, and thus may operate through common molecular grounds. This, to our knowledge, is a novel observation. We have provided further details about these insights in the revised manuscript.

3. The authors describe a phylum-specific AS event in the FGFR family required for normal developmental programs whose gene structure has evolved across metazoans. It is not really unexpected that ESRPs regulate this exon in other vertebrate species as well. The authors provide evidence for the evolution of flexible splicing programs in ESRP-dependent processes; however, it is at times unclear what the goals of many of the experiments are. Early on in the manuscript, it appears that the authors want to investigate the phenotype of ESRP DKO in Zebrafish. It was not until the end of the paper that the authors presented their work as a case for the evolution of flexible splicing programs. The study may be better organized under a single theme.

R: We apologize if the general goals of our study were not clear enough; we have now made them more explicit in the Abstract and in the last paragraph of the Introduction, as

well as throughout the manuscript. To clarify, our main goal was to investigate the functional evolution of *Esrp* and its associated regulated exon programs at different time-scales across deuterostomes. For this reason, we chose to study four different model organisms that are at increasing phylogenetic distances from mammals. As part of this, we found a high evolutionary turnover of the target programs between phyla, including the recruitment of an *Fgfr* AS event as an *Esrp* target in stem chordates. Given the crucial functional importance of this alternative splicing event for mouse *Esrp* KO's phenotypes, which we further expanded to zebrafish, we decided to focus on the evolution of this event to illustrate the origin of such flexibility.

Minor Comments:

1. The single mutant of *Esrp1* and *2* did not show developmental defects in comparison to DKO in Figure 1. Does the expression of the other ESRP paralogue increase as a compensating mechanism?

R: We thank the Reviewer for this suggestion. To address this question, we performed quantitative PCRs (qPCRs) for both genes in wild type, *esrp1* and *esrp2* single and double mutant embryos. We found that only the expression levels of the genes with the mutant alleles were strongly altered, while the other paralog maintains more similar levels to the WT. Thus compensatory mechanism is probably due to developmental robustness, rather than genetic expression compensation.

2. For Figure 3, some manipulation of ESRP expression in *Branchiostoma lanceolatum* should have been carried out to support the claim of conserved function. As it currently sits, it does not belong as main figure.

R: Unfortunately, experimental manipulations of *Branchiostoma lanceolatum* embryos are nearly impossible to perform, particularly within the time frame of this revision. However, we believe that expression patterns provide useful and suggestive information about the function of the gene in this organism, particularly when presented in an evolutionary context, and thus we would prefer to keep it as a main figure.

3. In figure 4, it is hard to see much of a difference between the control and morpholino-treated embryos between panels J and K. The legend for these panels and their reference in the text states that abnormally long cilia ("hairy" phenotype) can be seen in panel K, but it does not look like it. This is shown in panels N and O. For panel R, the authors do not offer an explanation of what differentiates between a strong or mild hairy phenotype.

R: We have added red arrowheads to panel J pointing at pigment cells that have already ingressed in the ectoderm in control gastrulas.

We apologize for the mistake in panel K description; it has been now corrected.

We have now provided a detailed explanation of the different levels of the phenotype in the figure legend. In particular, the defect is scored as "mild" when aboral ectoderm cells show long cilia (of length similar or longer than ciliary band cells), and "strong"

when, in addition to this, larvae show particularly long cilia at the apex, as indicated by an arrowhead in panel (o).

4. Also, in Figure 4, the authors use morpholinos to knockdown ESRP in sea urchin zygotes. However, it is unclear how intron 2 retention fails to produce functional ESRP protein. Also, why was translation inhibition used as a control for this knockdown?

R: Retention of intron 2 induces a premature termination codon (now depicted in Fig. 4p) that is predicted to induce non-sense mediated decay (NMD) and/or result in a highly truncated protein (containing only exons 1 and 2).

Translation inhibition with a second morpholino was not employed as a control, but to demonstrate the specificity of the knockdown by the use of two independent morpholino molecules against *Esrp* transcripts. We have now clarified this in the text.

5. The authors show in Figure 6c that zebrafish ESRP1 only produced mild changes in exon IIIb inclusion. Why is this the case? One would expect equivalent changes unless the presence of the IIIx exon is altering its splicing efficiency in some way. Please elaborate.

R: We also worked with the hypothesis that exon IIIx was altering the efficiency of zebrafish *Esrp1* to induce changes in the amphioxus FGFR splicing event. However, the experiment in which we remove exon IIIx from the minigene suggests that there have to be additional elements required for amphioxus-like regulation of the event by *Esrp* proteins. As the amphioxus protein is able to modify the isoform balance to a greater level, we speculated that zebrafish *Esrp1* protein is lacking specific characteristics “*in trans*” needed for proper interactions with the minigene RNA and/or components of the spliceosome. This could be related to an amphioxus-specific change at the protein level or a loss of undetermined ancient elements in all or some vertebrate’s orthologs.

6. Figure 7 shows the variability of the IgIII domain structure of FGFR, however, it does not show the evolutionary change of that structure through time as the authors suggest. Altering the figure in a way that clearly outlines the evolutionary timeline that the authors have explained in the text would enhance the readability of the manuscript.

R: We have now added explanatory information at specific points in the cladogram of Figure 7 to reconstruct the different steps inferred as the evolutionary scenario. Thank you for the suggestion.

7. Please explain why the RNAseq experiment on Zebrafish DKO and WT larvae was performed at 5 dpf, instead of before or later when survival rates go down?

R: In case of zebrafish, we selected animals at 5dpf because this was the earliest the double mutants could be unequivocally recognized by eye based on their phenotype. Use of earlier time points would imply fin-clipping hundreds of embryos at vulnerable stages (as double mutants had to be obtained through crosses of double heterozygous animals, with a 1:16 ratio). On the other hand, use of later stages would be easy, but presumably will result in the detection of more indirect effects due to more severe

developmental abnormalities. This rationale has now been included in the Methods section.

Also, was there a reason to choose SpIO-injected embryos, but not TrMo-injected embryos, for RNA-seq?

R: The splicing morpholinos was selected because we could test its efficiency by RT-PCR before sequencing (and corroborate it with the RNA-seq). It should be pointed out, however, that the observed phenotypes were very similar for both morpholino treatments.

8. In figure 2m-n, Twist>GFP may be a better control than Tyrp1/2a>2xGFP.

R: We had originally used this control to allow for better visualization of the results, given the well-established high efficiency of co-electroporation in this organism. However, we have now repeated the control electroporation experiment with Twist>GFP and Twist>Cherry, as suggested by the Reviewer, to show full co-expression of both plasmids (see Figure below). To make the image easier to understand for general readers, we nonetheless prefer to present only the Cherry channel in the main panel.

9. The language within the paper is confusing/difficult to follow at many places with a lot of unnecessary use of jargon. It also feels like the description of the results is stretched, making it unnecessarily lengthy. Therefore, the authors may want to modify and present results in a better and concise manner, changing figures to include pertinent information and improving the overall flow of the paper.

R: We have done our best to eliminate specific jargon and improve the general information flow of the manuscript. Requested additional analysis and experiments have been also added to figures. However, we believe that the manuscript was overall quite compact and we would not like to compromise its readability.

REVIEWERS' COMMENTS:

Reviewer #1 (Remarks to the Author):

The authors have replied to all my concerns in a satisfying way and performed all the changes.

Reviewer #2 (Remarks to the Author):

The authors have addressed all of my requests, and their new results are remarkable. I support its publication without any further changes needed.

I expect this study to be recognised a landmark in the field, and to be highly cited. It shows in the clearest way possible how the conserved role of specific splicing regulators explains the high conservation of intronic sequences around alternative exons. Only a few previous studies have addressed the evolution of splicing programmes, but none at such depth.

Reviewer #3 (Remarks to the Author):

I went through all the comments and their answers to it. It seems that compared to the last time the revised manuscript is much better now actually. And, in part, I believe the three reviewers comments have improved it.